# A decentralised neural model explaining optimal integration of navigational strategies in insects

**Xuelong Sun[1]\*, Shigang Yue[1,2]\*, Michael Mangan[3]\***

[1]Computational Intelligence Lab & L-CAS, School of Computer Science, University of Lincoln, Lincoln, United Kingdom; [2]Machine Life and Intelligence Research Centre, Guangzhou University, Guangzhou, China; [3]Sheffield Robotics, Department of Computer Science, University of Sheffield, Sheffield, United Kingdom

**Abstract** Insect navigation arises from the coordinated action of concurrent guidance systems but the neural mechanisms through which each functions, and are then coordinated, remains unknown. We propose that insects require distinct strategies to retrace familiar routes (route-following) and directly return from novel to familiar terrain (homing) using different aspects of frequency encoded views that are processed in different neural pathways. We also demonstrate how the Central Complex and Mushroom Bodies regions of the insect brain may work in tandem to coordinate the directional output of different guidance cues through a contextually switched ring-attractor inspired by neural recordings. The resultant unified model of insect navigation reproduces behavioural data from a series of cue conflict experiments in realistic animal environments and offers testable hypotheses of where and how insects process visual cues, utilise the different information that they provide and coordinate their outputs to achieve the adaptive behaviours observed in the wild.

**\*For correspondence:**
xsun@lincoln.ac.uk (XS);
syue@lincoln.ac.uk (SY);
m.mangan@sheffield.ac.uk (MM)

**Competing interests:** The authors declare that no competing interests exist.

## Introduction

Central-place foraging insects navigate using a 'toolkit' of independent guidance systems (*Wehner, 2009*) of which the most fundamental are path integration (PI), whereby foragers track the distance and direction to their nest by integrating the series of directions and distances travelled (for reviews see *Heinze et al., 2018*; *Collett, 2019*), and visual memory (VM), whereby foragers derive a homing signal by comparing the difference between current and stored views (for reviews see *Zeil, 2012*; *Collett et al., 2013*). Neurophysiological and computational modelling studies advocate the central complex neuropil (CX) as the PI centre (*Heinze and Homberg, 2007*; *Seelig and Jayaraman, 2015*; *Stone et al., 2017*), whereas the mushroom body neuropils (MB) appear well suited to assessing visual valence as needed for VM (*Heisenberg, 2003*; *Ardin et al., 2016*; *Müller et al., 2018*). Yet, two key gaps in our understanding remain. Firstly, although current VM models based on the MB architecture can replicate route following (RF) behaviours whereby insects visually recognise the direction previously travelled at the same position (*Ardin et al., 2016*; *Müller et al., 2018*), they cannot account for visual homing (VH) behaviours whereby insects return directly to their familiar surroundings from novel locations following a displacement (e.g. after being blown off course by a gust of wind) (*Wystrach et al., 2012*). Secondly, despite increasing neuroanatomical evidence suggesting that premotor regions of the CX coordinate navigation behaviour (*Pfeiffer and Homberg, 2014*; *Heinze and Pfeiffer, 2018*; *Honkanen et al., 2019*), a theoretical hypothesis explaining how this is achieved by the neural circuitry has yet to be developed. In this work, we present a unified neural navigation model that extends the core guidance modules from two (PI and VM) to three (PI,

RF, and VH) and by integrating their outputs optimally using a biologically realistic ring attractor network in the CX produces realistic homing behaviours.

The foremost challenge in realising this goal is to ensure that the core guidance subsystems provide sufficient directional information across conditions. Contemporary VM models based on the MBs can replicate realistic RF behaviours in complex visual environments (ant environments: *Kodzhabashev and Mangan, 2015*; *Ardin et al., 2016*, bee environments: *Müller et al., 2018*) but do not generalise to visual homing scenarios whereby the animal must return directly to familiar terrain from novel locations (ants: *Narendra, 2007*, bees: *Cartwright and Collett, 1982*, wasps: *Stürzl et al., 2016*). Storing multiple nest-facing views before foraging, inspired by observed learning walks in ants (*Müller and Wehner, 2010*; *Fleischmann et al., 2016*) and flights in bees and wasps (*Zeil et al., 1996*; *Zeil and Fleischmann, 2019*), provides a potential solution (*Graham et al., 2010*; *Wystrach et al., 2013*), but simulation studies have found this approach to be brittle due to high probabilities of aligning with the wrong memory causing catastrophic errors (*Dewar et al., 2014*). Moreover, ants released perpendicularly to their familiar route do not generally align with their familiar visual direction as predicted by the above algorithms (*Wystrach et al., 2012*), but instead move directly back towards the route (*Fukushi and Wehner, 2004*; *Kohler and Wehner, 2005*; *Narendra, 2007*; *Mangan and Webb, 2012*; *Wystrach et al., 2012*), which would require a multi-stage mental alignment of views for current models. New computational hypothesis are thus required that can guide insects directly back to their route (often moving perpendicularly to the habitual path), but also allow for the route direction to be recovered (now aligned with the habitual path) upon arrival at familiar surroundings (see *Figure 1A* 'Zero Vector').

With the necessary elemental guidance systems defined, a unifying model must then convert the various directional recommendations into a single motor command appropriate to the context (*Cruse and Wehner, 2011*; *Hoinville et al., 2012*; *Collett et al., 2013*; *Webb, 2019*). Behavioural studies show that when in unfamiliar visual surroundings ('Off-Route') insects combine the outputs of their PI and VH systems (*Collett, 1996*; *Bregy et al., 2008*; *Collett, 2012*) relative to their respective certainties consistent with optimal integration theory (*Legge et al., 2014*; *Wystrach et al., 2015*; *Figure 1A* 'Full Vector'). Upon encountering their familiar route, insects readily recognise their surroundings, recover their previous bearing and retrace their familiar path home (*Harrison et al., 1989*; *Kohler and Wehner, 2005*; *Wystrach et al., 2011*; *Mangan and Webb, 2012*). Thus, the navigation coordination model must posses two capabilities: (a) output a directional signal consistent with the optimal integration of PI and VH when Off-Route (b) switch from Off-Route (PI and VH) to On-Route (RF) strategies when familiar terrain is encountered. Mathematical models have been developed that reproduce aspects of cue integration in specific scenarios (*Cruse and Wehner, 2011*; *Hoinville and Wehner, 2018*), but to date no neurobiologically constrained network revealing how insects might realise these capabilities has been developed.

To address these questions a functional modelling approach is followed that extends the current base model described by *Webb, 2019* to (a) account for the ability of ants to home from novel locations back to the familiar route before retracing their familiar path the rest of the journey home, and (b) propose a neurally based model of the central complex neuropil that integrates competing cues optimally and generates a simple steering command that can drive behaviour directly. Performance is bench-marked by direct comparison to behavioural data reported by *Wystrach et al., 2012* (showing different navigation behaviours on and off the route), *Legge et al., 2014*; *Wystrach et al., 2015* (demonstrating optimal integration of PI and VM), and through qualitative comparison to extended homing paths where insects switch between strategies according to the context (*Narendra, 2007*). Biological realism is enforced by constraining models to the known anatomy of specific brain areas, but where no data exists an exploratory approach is taken to investigate the mechanisms that insects may exploit. *Figure 1A* depicts the adaptive behaviours observed in animals that we wish to replicate accompanied by a functional overview of our unified model of insect navigation (*Figure 1B*) mapped to specific neural sites (*Figure 1C*).

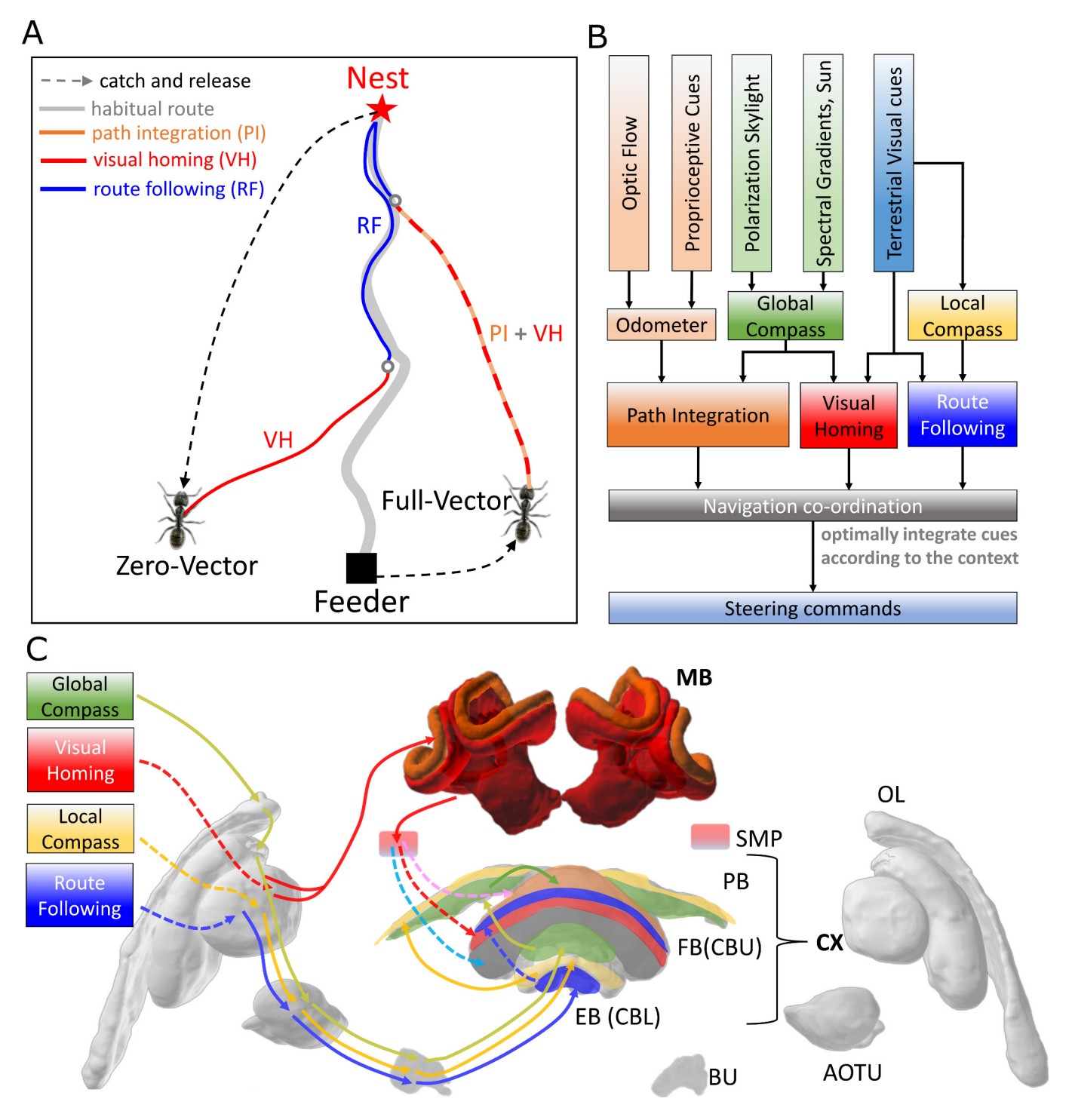

**Figure 1.** Overview of the unified navigation model and it's homing capabilities. (**A**) The homing behaviours to be produced by the model when displaced either from the nest and having no remaining PI home vector (zero vector), or from the nest with a full home vector (full vector). Distinct elemental behaviours are distinguished by coloured path segments, and stripped bands indicate periods where behavioural data suggests that multiple strategies are combined. Note that this colour coding of behaviour is maintained throughout the remaining figures to help the reader map function to brain region. (**B**) The proposed conceptual model of the insect navigation toolkit from sensory input to motor output. Three elemental guidance systems are modelled in this paper: path integration (PI), visual homing (VH) and route following (RF). Their outputs must then be coordinated in an optimal manner appropriate to the context before finally outputting steering command. (**C**) The unified navigation model maps the elemental guidance systems to distinct processing pathways: *RF*: OL - > AOTU - > BU - > CX; *VH*: OL - > MB - > SMP - > CX; *PI*: OL - > AOTU - > BU - > CX. The outputs

*Figure 1 continued on next page*

*Figure 1 continued*

are then optimally integrated in the proposed ring attractor networks of the FB in CX to generate a single motor steering command. Connections are shown only for the left brain hemisphere for ease of visualisation but in practice are mirrored on both hemispheres. Hypothesised or assumed pathways are indicated by dashed lines whereas neuroanatomically supported pathways are shown by solid lines (a convention maintained throughout all figures). *OL*: optic lobe, *AOTU*: anterior optic tubercle, *CX*: central complex, *PB*: protocerebrum bridge, *FB*: fan-shape body (or *CBU*: central body upper), *EB*: ellipsoid body (or *CBL*: central body lower), *MB*: mushroom body, *SMP*: superior medial protocerebrum, *BU*: bulb. Images of the brain regions are adapted from the insect brain database https://www.insectbraindb.org.

## Results

### Mushroom bodies as drivers of rotational invariant visual homing

For ants to return directly to their familiar route after a sideways displacement (*Figure 1A* 'Zero Vector') without continuous mental or physical realignment they require access to rotational invariant visual cues. *Stone et al., 2018* recently demonstrated that binary images of panoramic skylines converted into their frequency components can provide such a rotationally-invariant encoding of scenes in a compact form (see Image processing for an introduction to frequency transformations of images). Moreover, they demonstrated that the difference between the rotationally invariant features (the amplitudes of the frequency coefficients) between two locations increases monotonically with distance producing an error surface reminiscent of the image difference surfaces reported by *Zeil et al., 2003* which can guide an agent back to familiar terrain. Here we investigate whether the MB neuropils shown capable of assessing the visual valence of learned rotationally-varying panoramic skylines for RF (*Ardin et al., 2016*; *Müller et al., 2018*), might instead assess the visual valence of rotationally-invariant properties of views sampled along a familiar route supporting visual homing.

To this end, the intensity sensitive input neurons of *Ardin et al., 2016*'s MB model are replaced with input neurons encoding rotational invariant amplitudes (*Figure 2A* left, blue panel). The network is trained along an $11m$ curved route in a simulated world that mimics the training regime of ants in *Wystrach et al., 2012* (see Materials and methods and Reproduce visual navigation behaviour for details on simulated world, image processing, model architecture and training and test regime). After training, the firing rate of the MB output neuron (MBON) when placed at locations across the environment at random orientations reveals a gradient that increases monotonically with distance from the familiar route area, providing a homing signal sufficient for VH independent of the animal's orientation (*Figure 2C*).

Motor output is then generated by connecting the MBON to a steering network recently located in the fan-shaped body (FB/CBU) of the CX that functions by minimising the difference between the animal's current and desired headings (*Stone et al., 2017*). *Stone et al., 2017*'s key insight was that the anatomically observed shifts of activity in the columnar neurons that encode the desired heading in essence simulate 45° turns left and right, and thus by comparing the summed differences between the activity profiles of these predicted headings to the current heading then the appropriate turning command can be computed (see *Figure 2B*). We adopt this circuit as the basis for computing steering commands for all strategies as suggested by *Honkanen et al., 2019*.

In the proposed VH model the current heading input to the steering circuit uses the same celestial global compass used in *Stone et al., 2017*'s PI model. Insects track their orientation through head-direction cells *Seelig and Jayaraman, 2015* whose concurrent firing pattern forms a single bump of activity that shifts around the ring as the animal turns (measured through local visual [*Green et al., 2017*; *Turner-Evans et al., 2017*], global visual (*Heinze and Homberg, 2007*) and proprioceptive (*Seelig and Jayaraman, 2015*) cues). Neuroanatomical data (*Kim et al., 2017*; *Turner-Evans et al., 2019*; *Pisokas et al., 2019*) supports theoretical predictions (*Cope et al., 2017*; *Kakaria and de Bivort, 2017*) that the head-direction system of insects follows a ring attractor (RA) connectivity pattern characterised by local excitatory interconnections between direction selective neurons and global inhibition. In this work, the global compass RA network is not modelled directly but rather we simulate its sinusoidal activity profile in a ring of I-TB1 (locusts and $\Delta7$ of flies) neurons found in the protocerebral bridge (PCB/PB) (*Figure 2A* green ring) (see Current headings).

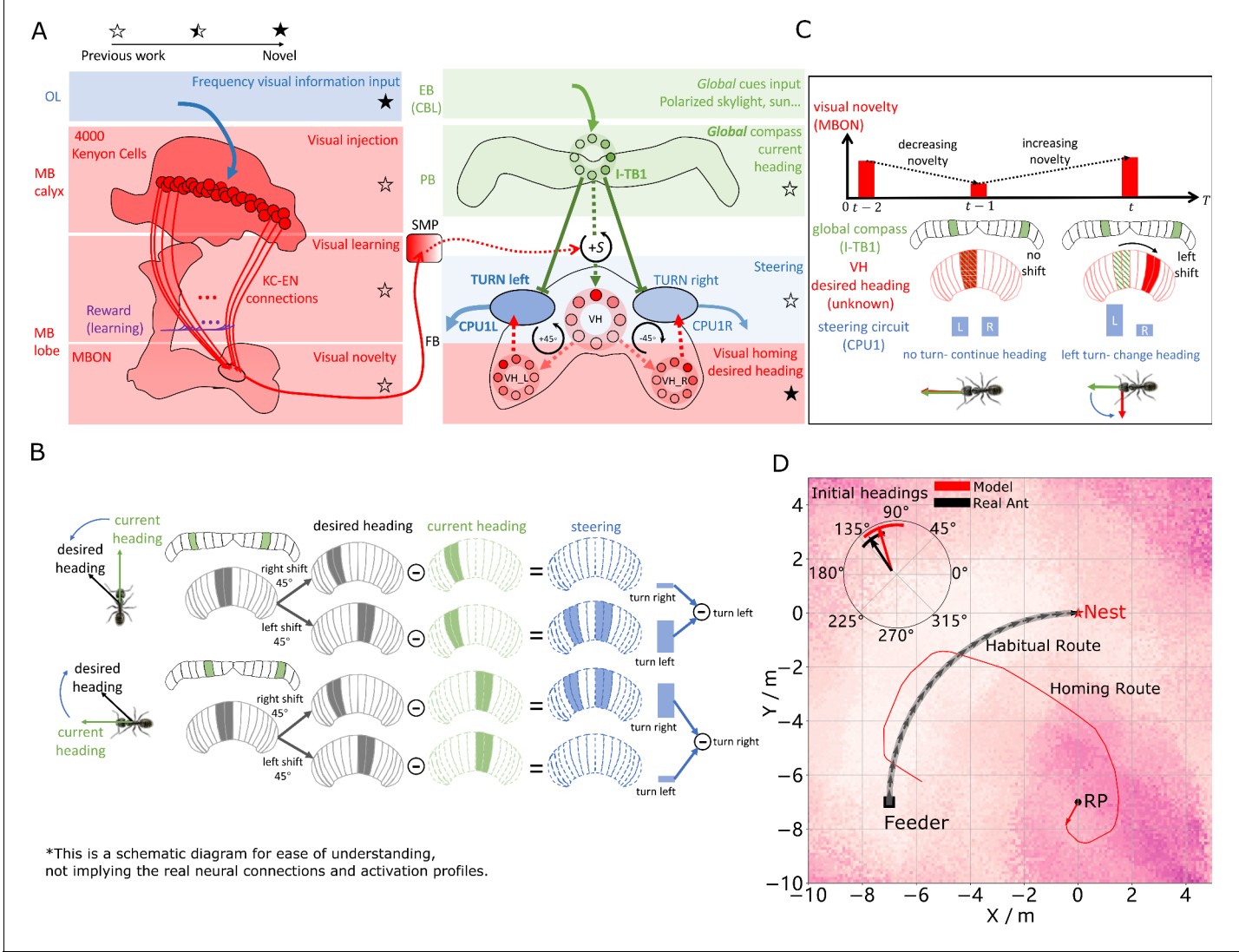

**Figure 2.** Visual homing in the insect brain. (**A**) Neural model of visual homing. Rotational-invariant amplitudes are input to the MB calyx which are then projected to the Kenyon cells (KCs) before convergence onto the MB output neuron (MBON) which seeks to memorise the presented data via reinforcement-learning-based plasticity (for more details see Visual homing) (MB circuit: left panels). SMP neurons measure positive increases in visual novelty (through input from the MBON) which causes a shift between the current heading (green cells) and desired headings (red cells) in the rings of the CX (SMP pathway between MB and CX: centre panel; CX circuit: right panels). The CX-based steering circuit then computes the relevant turning angle. Example activity profiles are shown for an increase in visual novelty, causing a shift in desired heading and a command to change direction. Each model component in all figures is labelled with a shaded star to indicate what aspects are new versus those incorporated from previous models (see legend in upper left). (**B**) Schematic of the steering circuit function. First the summed differences between the impact of 45 °left and right turns on the desired heading and the current heading are computed. By comparing the difference between the resultant activity profiles allows an appropriate steering command to be generated. (**C**) Schematic of the visual homing model. When visual novelty drops ($t − 2$ to $t − 1$) the desired heading is an unshifted copy of the current heading so the current path is maintained but when the visual novelty increases ($t − 1$ to $t$) the desired heading is shifted from the current heading. (**D**) The firing rate of the MBON sampled across locations at random orientations is depicted by the heat-map showing a clear gradient leading back to the route. The grey curve shows the habitual route along which ants were trained. RP (release point) indicates the position where real ants in *Wystrach et al., 2012* were released after capture at the nest (thus zero-vector) and from which simulations were started. The ability of the VH model to generate realistic homing data is shown by the initial paths of simulated ants which closely match those of real ants (see inserted polar plot showing the mean direction and 95% confidential interval), and also the extended exampled path shown (red line). Note that once the agent arrives in the vicinity of the route, it appears to meander due the flattening of visual novelty gradient and the lack of directional information. The online version of this article includes the following source data for figure 2:

**Source data 1.** The frequency information for the locations with random orientations across the world.
**Source data 2.** The visual homing results of the model.

A desired heading is then generated by copying the current activity pattern of the global compass neurons to a new neural ring which we speculate could reside in either a distinct subset of I-TB1 neurons (*Beetz et al., 2015*) or in the FB. Crucially, the copied activity profile also undergoes a leftward shift proportional to any increase in visual novelty (a similar shifting mechanisms has been proposed for the head-direction system [*Green et al., 2017*; *Turner-Evans et al., 2017*]) which we propose is measured by neurons in the superior medial protocerebrum (SMP) (*Aso et al., 2014*; *Plath et al., 2017*) (see *Figure 2A* centre and activity of red rings). The result is a mechanism that recommends changing direction when the agent moves away from familiar terrain (visual novelty increases) but recommends little change to the current heading when the visual novelty is decreasing (see *Figure 2C* for a schematic of the VH mechanism). We note that there is a distinction between a ring network which describes a group of neurons whose pattern of activity forms a circular representation regardless of actual physical arrangement and RA networks which follow a specific connectivity pattern (all modelled RAs labelled in figures). Taken together the model iteratively refines it's orientation to descend the visual novelty gradient and thus recover familiar terrain (see *Figure 2A* for full model).

*Figure 2D* demonstrates that the proposed network accurately replicates both the directed initial paths as in *Wystrach et al., 2012* (see the inserted black arrow), and extended homing paths as in *Narendra, 2007* observed in ants displaced to novel locations perpendicular to their familiar routes. We note that upon encountering the route the model is unable to distinguish the direction in which to travel and thus meanders back and forth along the familiarity valley, unlike real ants, demonstrating the need for additional route recognition and recovery capabilities.

## Optimally integrating visual homing and path integration

We have demonstrated how ants could use visual cues to return to the route in the absence of PI but in most natural scenarios (e.g. displacement by a gust of wind) ants will retain a home vector readout offering an alternative, and often conflicting, guidance cue to that provided by VH. In such scenarios, desert ants strike a comprise by integrating their PI and VH outputs in a manner consistent with optimal integration theory by weighting VH relative to the familiarity of the current view (*Legge et al., 2014*) and PI relative to the home vector length (a proxy for directional certainty) (*Wystrach et al., 2015*).

Various ring-like structures of the CX represent directional cues as bumps of activity with the peak defining the specific target direction, and the spread providing a mechanism to encode cue certainty as required for optimal integration (for an example see increased spread of HD cell activity when only proprioceptive cues are present [*Seelig and Jayaraman, 2015*]). Besides their excellent properties to encode the animal's heading ring attractors also provide a biologically realistic means to optimally weight cues represented in this format (*Touretzky, 2005*; *Mangan and Yue, 2018*) without the need for dedicated memory circuits to store means and uncertainties of each cue.

Thus we introduce a pair of integrating ring-attractor networks to the CX model (*Figure 3A* grey neural rings: RA_L and RA_R) that take as input the desired headings from the above proposed VH model (red neural rings: VH_L and VH_R) and *Stone et al., 2017*'s PI model (orange neural rings: PI_L and PI_R) and output combined Off Route desired heading signals that are sent to the steering circuits (blue neural rings: CPU_L and CPU_R). *Stone et al., 2017* mapped the home vector computation to a population of neurons (CPU4) owing to their dual inputs from direction selective compass neurons (I_TB1) and motion-sensitive speed neurons (TN2) as well as their recurrent connectivity patterns facilitating accumulation of activity as the animal moves in a given direction. *Wystrach et al., 2015* showed that the certainty of PI automatically scales with the home-vector length owing to the accumulating effect of the memory neurons which correlates with directional uncertainty, and thus the output PI network is directly input to the ring attractor circuits. In our implementation the VH input has a fixed height and width profile and influences the integration through tuning neurons (TUN) (see the plotted activation function in *Figure 3B* and Optimal cue integration) that we suggest reside in the SMP and modulate the PI input to the integration network. Altering the weighting in this manner rather than by scaling the VH input independently allows VH to dominate the integrated output at sites with high visual familiarity even in the presence of a large home vector without having large stored activity. We note, however, that both approaches remain feasible and further neuroanatomical data is required to clarify which, if either, mechanism is employed by insects.

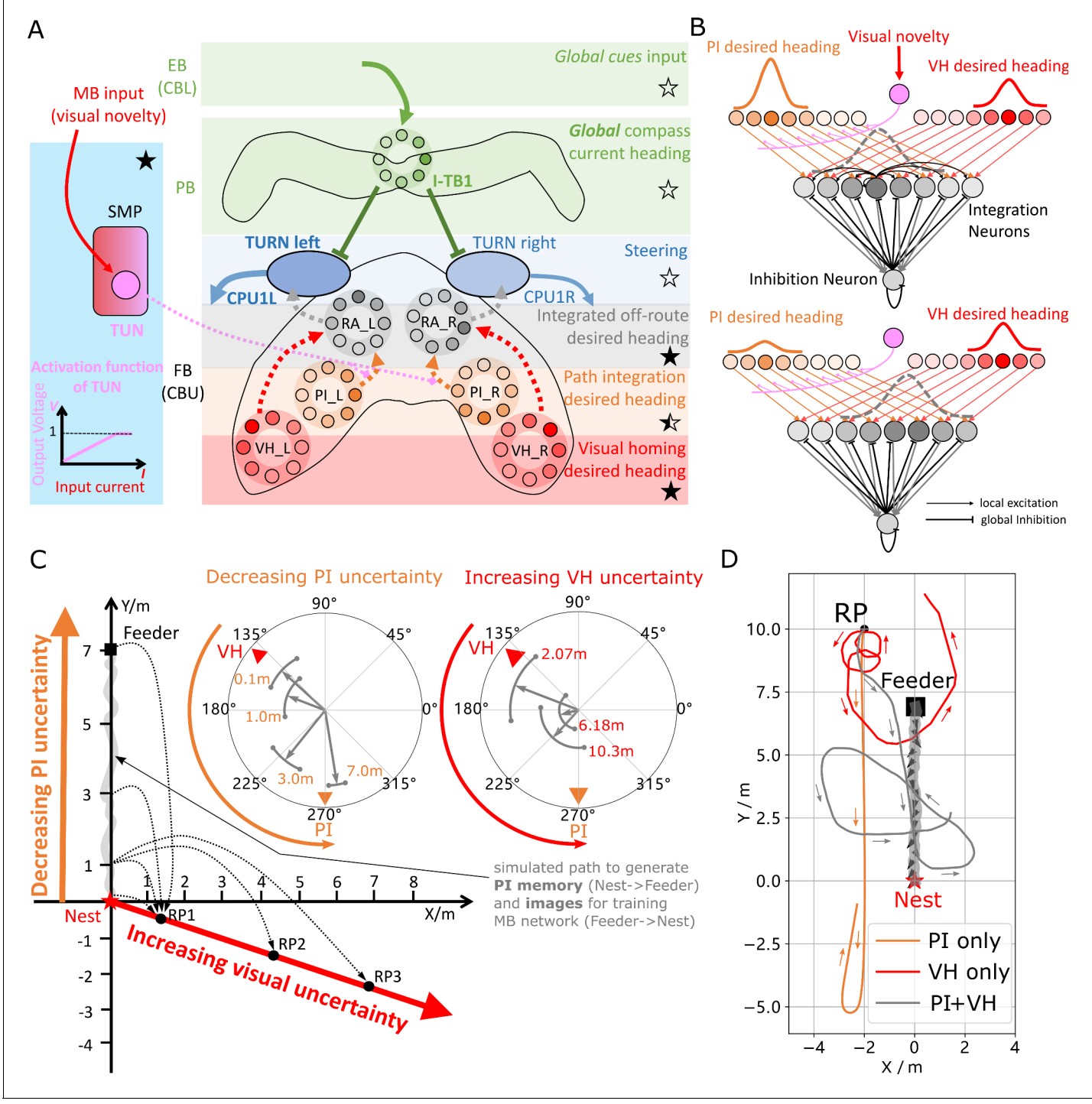

**Figure 3.** Optimal cue integration in the CX. (A) Proposed model for optimally integrating PI and VH guidance systems. In each hemisphere, ring attractors (RAs) (grey neural rings) (speculatively located in FB/CBU) receive the corresponding inputs from PI (orange neural rings) and VH (red neural rings) with the outputs sent to the corresponding steering circuits (blue neural rings). Integration is weighted by the visual novelty tracking tuning neuron (TUN) whose activation function is shown in the leftmost panel. (B) Examples of optimal integration of PI and VH headings for two PI states with the peak stable state (grey dotted activity profile in the integration neurons) shifting towards VH as the home vector length recedes. (C) Replication of optimal integration studies of *Wystrach et al., 2015* and *Legge et al., 2014*. Simulated ants are captured at various points (0.1 m, 1 m, 3 m and 7 m) along their familiar route (grey curve) and released at release point 1 (RP1) thus with the same visual certainty but with different PI certainties as in *Wystrach et al., 2015* (see thick orange arrow). The left polar plot shows the initial headings of simulated ants increasingly weight their PI system (270°) in favour of their VH system (135°) as the home vector length increases and PI directional uncertainty drops. Simulated ants are also transferred from a

*Figure 3 continued on next page*

*Figure 3 continued*

single point 1 m along their familiar route to ever distant release points (RP1, RP2, RP3) thus with the same PI certainty but increasingly visual uncertainty as in *Legge et al., 2014* (see thick red arrow). The right polar plot shows the initial headings of simulated ants increasingly weight PI (270˚) over VH (135˚) as visual certainty drops. (see Reproduce the optimal cue integration behaviour for details) (D) Example homing paths of the independent and combined guidance systems displaced from the familiar route (grey) to a fictive release point (RP).

The online version of this article includes the following source data and figure supplement(s) for figure 3:

**Source data 1.** The results of tuning PI uncertainty.
**Source data 2.** The results of tuning VH uncertainty.
**Source data 3.** The extended homing path of PI, VH and combined PI and VH.
**Figure supplement 1.** The extended homing paths and the PI memory in the simulations.

*Figure 3C* shows the initial headings produced by the model which replicates the trends reported in cue-conflict experiments by *Legge et al., 2014* and *Wystrach et al., 2015* when the uncertainty of PI and VH cues were altered independently. Example extended paths of independent PI and VH models and the ring-attractor-based combined PI and VH model are plotted in *Figure 3D* with the combined model showing the most ant-like behaviour (*Kohler and Wehner, 2005*; *Mangan and Webb, 2012*) by initially following predominantly the home-vector direction before switching to visual homing when the home-vector length drops leading the simulated ant back to familiar terrain. Note that the PI-only and PI+VH models are drawn back towards their fictive nest sites indicated by their home vectors which if left to run would likely result in emergent search-like patterns as in *Stone et al., 2017*. Moreover, upon encountering the route the VH-based models (VH-only and PI +VH) are unable to distinguish the direction in which to travel and hence again (see meander around the valley of familiarity *Figure 2D* and *Figure 3D*) further demonstrating a need for a route recovery mechanism.

## Route following in the insect brain

The model described above can guide insects back to their familiar route area, but lacks the means to recover the route direction upon arrival as observed in homing insects. This is not surprisingly as VH relies upon translationally-varying but rotational-invariant information whereas RF requires rotationally varying cues. Thus we introduce a new elemental guidance system that makes use of the rotationally-varying *phase* coefficients of the frequency information derived from the panoramic sky-line which tracks the orientation of specific features of the visual surroundings (see Materials and methods). Here, we ask whether by associating the rotationally invariant amplitudes (shown useful for place recognition) with the rotationally-varying *phases* experienced at those locations, insects might recover the familiar route direction.

Neuroanatomical data with which to constrain a model remains sparse and therefore a standard artificial neural network (ANN) architecture is used to investigate the utility of *phase*-based route recovery with biological plausibility discussed in more detail below. A three-layer ANN was trained to associate the same 81 rotational-invariant amplitudes as used in the VH model with the rotational varying *phase* value of single frequency coefficient experienced when travelling along the habitual route which we encode in an eight neuron-ring (see *Figure 4A* and Route Following for detailed model description). Thus, when the route is revisited the network should output the orientation that the *phase* converged upon when at the same location previously, which we note is not necessarily aligned with the actual heading of the animal (e.g. it may track the orientation to vertical bar [*Seelig and Jayaraman, 2015*]). Realignment is possible using the same steering mechanism as described above but which seeks to reduce the offset between the current *phase* readout (e.g. a local compass locked onto visual features of the animals surroundings), and the recalled *phase* read-out from the ANN.

We speculate that the most likely neural pathways for the new desired and current headings are from Optic Lobe via Anterior Optic Tubercle (AOTU) and Bulb (BU) to EB (CBL) of the CX (*Homberg et al., 2003*; *Omoto et al., 2017*) (see *Figure 4A*) with the desired heading terminating in the EB, whereas the current heading continues to the PB forming a local compass that sits beside the global compass used by PI and VH systems. This hypothesis is further supported by the recently identified parallel pathways from OL via AOTU to the CX in *Drosophila* (*Timaeus et al., 2020*). That's to say that, firstly, there are two parallel pathways forming two compass systems- the global (here

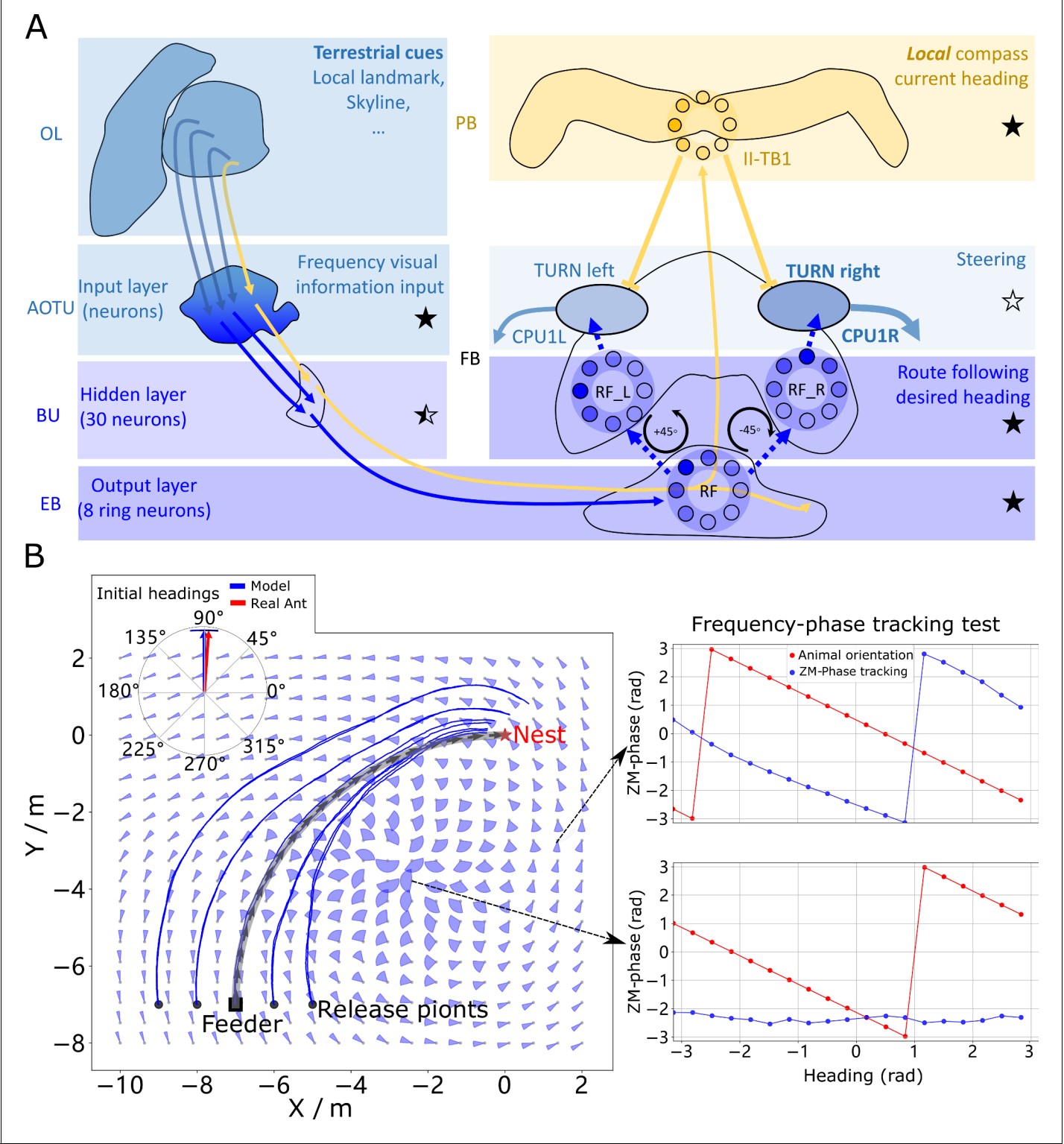

**Figure 4.** Phase-based route following. (**A**) Neural model. The visual pathway from the optic lobe via AOTU and Bulb to EB of the CX is modelled by a fully connected artificial neural network (ANN) with one hidden layer. The input layer receives the amplitudes of the frequency encoded views (as for the MB network) and the output layer is an 8-neuron ring whose population encoding represents the desired heading against to which the agent should align. (**B**) Behaviours. Blue and red arrows in the inserted polar plot (top left) display the mean directions and 95% confidential intervals of the initial headings of real (*Wystrach et al., 2012*) and simulated ants released at the start of the route (−7, −7), respectively. Dark blue curves show the routes followed by the model when released at five locations close to the start of the learned path. The overlaid fan-plots indicate the circular statistics (the

*Figure 4 continued on next page*

*Figure 4 continued*

mean direction and 95% confidential interval) of the homing directions recommended by the model when sampled across heading directions (20 samples at 18°intervals). Data for entire rotations are shown on the right for specific locations with the upper plot, sampled at (1.5, −3), demonstrating accurate phase-based tracking of orientation, whereas the lower plot sampled at (−2.5, −3.5) shows poor tracking performance and hence produces a wide fan-plot.

The online version of this article includes the following source data for figure 4:

**Source data 1.** The frequency tracking performance across the world.
**Source data 2.** The RF model results of the agents released on route.
**Source data 3.** The RF model results of the agents released aside from the route.

based on celestial cues) and the local (based on terrestrial cues) compasses modelled by the activation of l-TB1 and ll-TB1 neurons, respectively. Four classes of CL1 neurons (or E-PG and P-EG neurons) *Heinze and Homberg, 2009*; *Xu et al., 2020* and three classes of independent TB1 neurons *Beetz et al., 2015* have been identified that provide potential sites for the parallel recurrent loops encoding independent local and global compasses. Secondly, the desired heading, which is the recalled *phase* of a specific view, is generated through the neural plasticity from AOTU to BU and BU to EB, which is line with recent evidence of associative learning between the R-neurons transmitting visual information from BU to EB and the compass neurons (CL1a or E-PG neurons) that receive input from EB (*Kim et al., 2019*; *Fisher et al., 2019*). This kind of learning endows the animal with the ability to flexibly adapt their local compass and also desired navigational orientation according to the changing visual surroundings. *Hanesch et al., 1989* reported a direct pathway from EB to FB neurons which we model to allow comparison of the local compass activity (ll-TB1) with the desired heading. However, we note that this connectivity has not been replicated in recent studies *Heinze and Homberg, 2008* and thus further investigation of potential pathways is required.

The RF model accurately recovers the initial route heading in a similar manner to real ants returned to the start of their familiar route (*Wystrach et al., 2012*; *Figure 4B*, insert), and then follows the remaining route in its entirety back to the nest again reflecting ant data (*Kohler and Wehner, 2005*; *Mangan and Webb, 2012*; *Figure 4B*). The quiver plots displayed in the background of *Figure 4B* show the preferred homing direction output by the ANN when rotated on the spot across locations in the environment. The noise in the results are due to errors in the tracking performance (see examples *Figure 4B* right) yet as these errors are in largely confined to the magnitude, the steering circuit still drives the ant along the route. We note that this effect is primarily a function of the specific frequency transformation algorithm used which we borrow from computer graphics to investigate the utility of frequency encoding of visual information. The biological realism of such transforms and their potential implementation in the insect vision system are addressed in the Discussion. The displaced routes also highlight the danger of employing solely RF which often shadows rather than converges with the route when displaced sideways, further demonstrating the necessity for integration with the Off-Route strategies that promote route convergence.

## Route recovery through context-dependent modulation of guidance systems

Homing insects readily recognise familiar route surroundings, recover their bearing, and retrace their habitual path home, irrespective of the status of other guidance system such as PI. Replicating such context-dependent behavioural switching under realistic conditions is the final task for the proposed model. The visual novelty measured by the MBON provides an ideal signal for context switching with low output when close to the route when RF should dominate versus high output further away from the route when PI and VH should be engaged (see *Figure 2D*). Also the fact that Off-route strategies (PI and VH) compute their turning angles with reference to the global compass whereas the On-route RF strategy is driven with reference to a local compass provides a means to modulate their inputs to the steering circuit independently. This is realised through a non-linear weighting of the On and Off-route strategies which we propose acts through the same SMP pathway as the VH model (see the SN1 and SN2 neurons in *Figure 5A*) (see Context-dependent switch for neuron details and *Figure 6* for a force-directed graph representation of the final unified model).

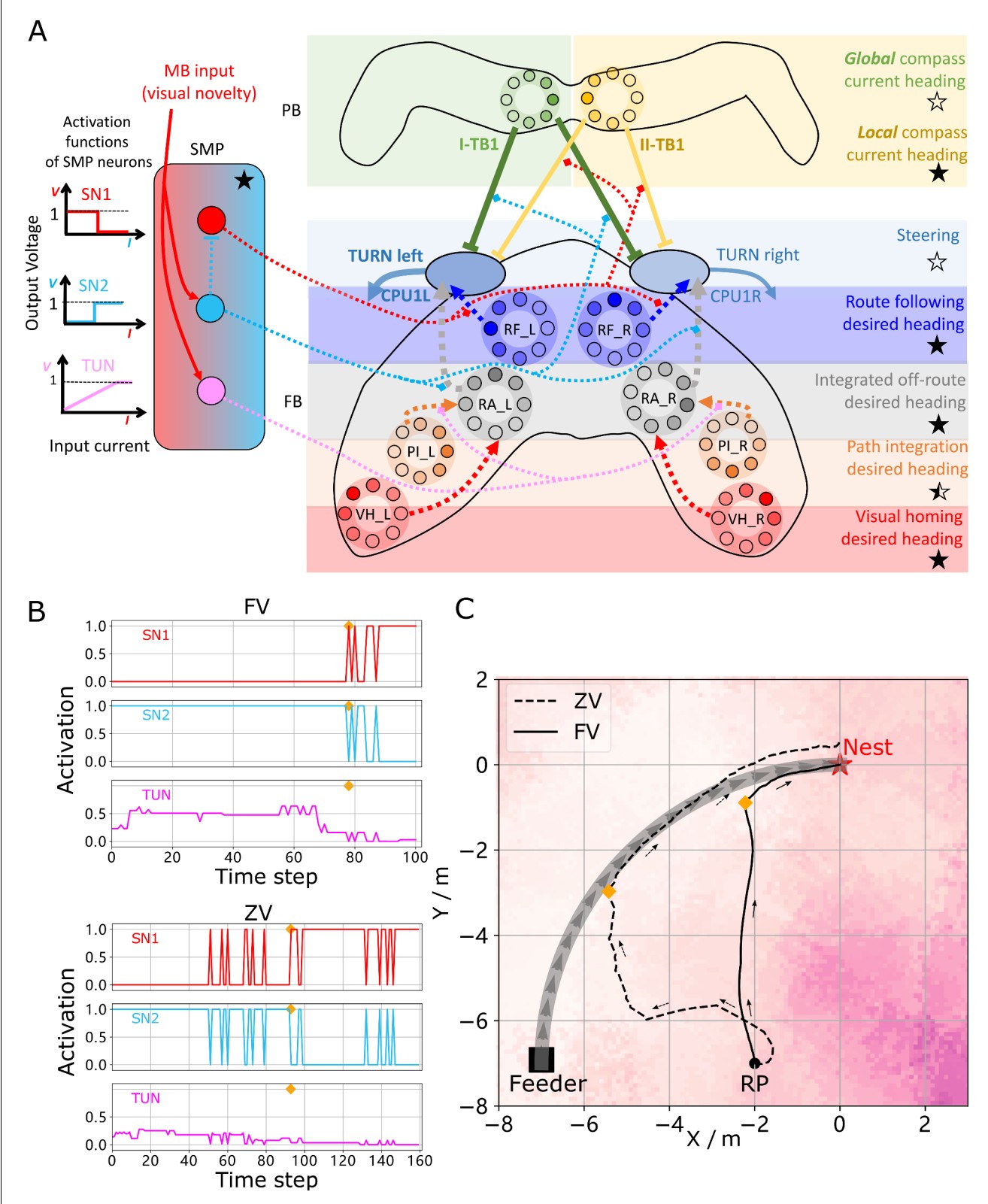

**Figure 5.** Unified model realising the full array of coordinated navigational behaviours. (**A**) Context-dependent switching is realised using two switching neurons (SN1, SN2) that have mutually exclusive firing states (one active while the other is in active) allowing coordination between On and Off-Route strategies driven by the instantaneous visual novelty output by the MB. Connectivity and activation functions of the SMP neurons are shown in the left side of panel. (**B**) Activation history of the SN1, SN2 and TUN (to demonstrate the instantaneous visual novelty readout of the MB) neurons during the

*Figure 5 continued on next page*

*Figure 5 continued*

simulated displacement trials. (**C**) Paths generated by the unified model under control of the context-dependent switch circuit during simulated FV (solid line) and ZV (dashed line) displacement trials.

The online version of this article includes the following source data for figure 5:

**Source data 1.** The navigation results of the whole model.

The activity of the proposed switching circuit and the paths that it generates in simulated zero vector and full vector displacement trials are shown in *Figure 5B and C* respectively. In the full vector trial (*Figure 5B* (upper), *Figure 5C* (solid line)) as visual novelty is initially high (see high TUN activity until step 78) SN2 is activated which enables Off-Route strategies (PI and VH) while SN1 (always the inverse of SN2) is deactivated which disables On-Route strategies. Note that it is the integration of PI and VH that generates the direct path back to the route area in the FV trial: PI recommends moving at a 45° bearing but VH prevents ascension of the visual novelty gradient that this would cause with the compromise being a bearing closer to 90° that is toward the route. As the route is approached the visual novelty decreases (again see TUN activity), until at step 78 SN2 falls below threshold and deactivates the Off-Route strategies while conversely SN1 activates and engages On-Route strategies. After some initial flip-flopping while the agents converges on the route (steps 78–85) RF becomes dominant and drives the agent back to the nest via the familiar path. In the zero vector trial (*Figure 5B* (lower), (*Figure 5B* (dashed line)) Off-route strategies (here only VH) largely dominate (some false positive route recognition (e.g step 60)) until the route is recovered (step 93), at which point the same flip-flopping during route convergence occurs (steps 93–96) followed by RF alone which returns the agent to the nest via the familiar path. It should be noted that the data presented utilised different activation functions of the TUN neuron that weights PI and VH (see *Table 1* for parameter settings across trials and Discussion for insights into model limitations and potential extensions), yet the results presented nevertheless provide a proof-of-principle demonstration that the proposed unified navigation model can fulfil all of the criteria defined for replication of key adaptive behaviour observed in insects (*Figure 1A*).

## Discussion

This work addresses two gaps in the current understanding of insect navigation: what are the core visual guidance systems required by the insect navigational toolkit? And how are they coordinated by the insect brain?

We propose that the insect navigation toolkit (*Wehner, 2009*; *Webb, 2019*) should be extended to include independent visual homing (VH) and route following (RF) systems (see *Figure 1B* for updated Insect Navigation Toolkit). We show how VH and RF can be realised using frequency-encoding of panoramic skylines to separate information into rotationally invariant amplitudes for VH and rotationally varying *phases* for RF. The current model utilises frequency encoding schema from the computer graphics but behavioural studies support the use of spatial frequency by bees (*Horridge, 1997*; *Lehrer, 1999*), with neurons in the lobula of dragonflies (*O'Carroll, 1993*) and locusts *James and Osorio, 1996* found to have receptive fields akin to basis functions, providing a mechanism by which to extract the frequency information necessary for the local compass system. Our model allows for this information extraction process to happen at multiple stages ahead of its usage in the central learning sites such as the MBs opening the possibility for its application in either the optic lobes or subsequent pathways through regions such as the AOTU. Further, neurophysiological data is required to pinpoint both the mechanisms and sites of this data processing in insects. Similarly, following *Stone et al., 2017* the global compass signal directly mimics the firing pattern of compass neurons in the CX without reference to sensory input but *Gkanias et al., 2019* recently presented a plausible neural model of the celestial compass processing pipeline that could be easily integrated into the current model to fill this gap. Follow-on neuroanatomically constrained modelling of the optic lobes presents the most obvious extension of this work allowing the neural pathway from sensory input to motor output signal to be mapped in detail. Conversely, modelling the conversion of direction signals into behaviour via motor generating mechanisms such as central pattern generators (see *Steinbeck et al., 2020*) will then allow closure of the sensory-motor loop.

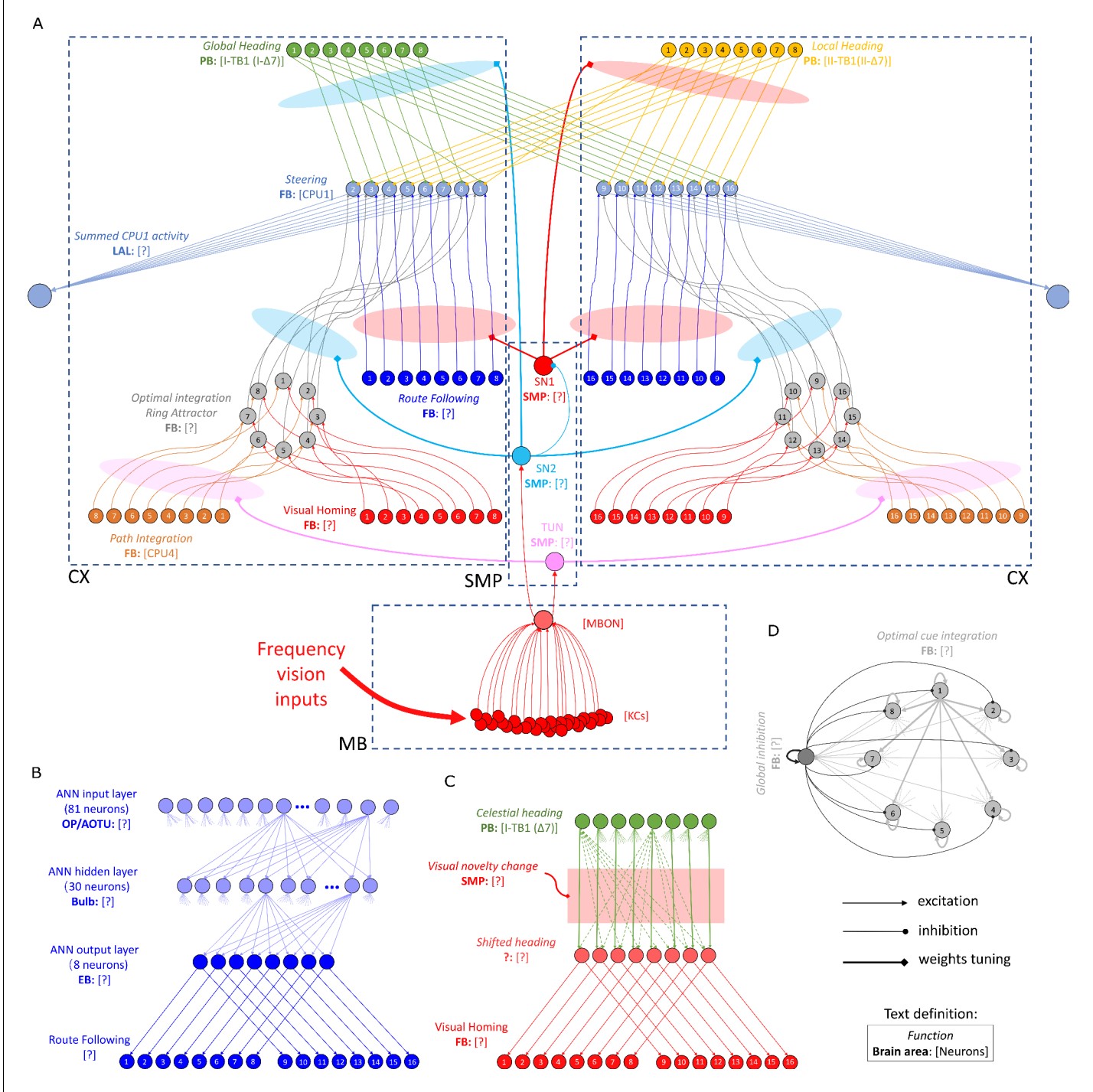

**Figure 6.** The detailed neural connections of the proposed model. (**A**): The detailed neural connections of the navigation coordination system. (**B**): The neural connection of the route following network. The input layer to the hidden layer is fully connected, so does the hidden layer to the output layer. (**C**): The network generating the visual homing memory. (**D**): The detailed neural connection of the ring attractor network for optimal cue integration.

Visual homing is modelled on neural circuits found along the OL-MB-SMP pathway (*Ehmer and Gronenberg, 2002*; *Gronenberg and López-Riquelme, 2004*) before terminating in the CX steering circuit (*Stone et al., 2017*) and shown capable of producing realistic homing paths. In this schema, the MBs do not measure rotationally varying sensory valence as recently used to replicate RF (*Ardin et al., 2016*; *Müller et al., 2018*), but rather the spatially varying (but rotationally invariant)

**Table 1.** The detailed parameters settings for the simulations.

| Para. | Visual homing | Optimal integration tuning PI | Optimal integration tuning VH | Route following | Whole model ZV | Whole model FV |
|---|---|---|---|---|---|---|
| $Thr_{KC}$(**Equation 14**) | 0.04 | 0.04 | 0.04 | 0.04 | 0.04 | 0.04 |
| $\eta_{KC2MBON}$(**Equation 16**) | 0.1 | 0.1 | 0.1 | 0.1 | 0.1 | 0.1 |
| $k_{VH}$(**Equation 19**) | 2.0 | 2.0 | 2.0 | / | 0.5 | 0.5 |
| $k_{TUN}$ (**Equation 28**) | / | 0.1 | 0.1 | / | 0.025 | 0.0125 |
| $Thr_{SN2}$(**Equation 32**) | / | / | / | / | 2.0 | 3.0 |
| $k_{motor}$(**Equation 35**) | 0.125 | 0.125 | 0.125 | 0.125 | 0.375 | 0.375 |
| $S_L$ (cm/step) (**Equation 39**) | 4 | 4 | 4 | 4 | 8 | 8 |
| initial heading (deg) | 0~360 | 0~360 | 0~360 | 0 / 180 | 90 | 0 |

sensory valence more suited to gradient descent strategies such as visual homing (**Zeil et al., 2003**; **Stone et al., 2018**) and other taxis behaviours (**Wystrach et al., 2016**). This is inline with the hypothesis forwarded by **Collett and Collett, 2018** that suggest that the MBs output 'whether' the current sensory stimulus is positive or negative and the CX then adapts the animal heading, the 'whither', accordingly.

Route following is shown possible by learned associations between the amplitudes (i.e. the place) and the *phase* (the orientation) experienced along a route, allowing realignment when later at a proximal location. This kind of neural plasticity-based correlation between the visual surroundings and the orientations fits with data recently observed in fruit flies (**Kim et al., 2019**; **Fisher et al., 2019**). These studies provide the neural explanation for the animal's ability to make flexible use of visual information to navigate while the proposed model gives a detailed implementation of such ability in the context of insect's route following schema. Neurophysiological evidence suggests that the layered visual pathway from OL via AOTU and BU to the EB of the CX (**Barth and Heisenberg, 1997**; **Homberg et al., 2003**; **Omoto et al., 2017**) with its suggested neural plasticity properties (**Barth and Heisenberg, 1997**; **Yilmaz et al., 2019**) provides a possible neural pathway but further analysis is needed to identify the circuit structures that might underpin the generation of RF desired heading. In addition to the desired heading, the current heading of RF is derived from the local compass system anchored to animal's immediate visual surroundings. This independent compass system may be realised parallel to the global compass system in an similar but independent circuit (**Heinze and Homberg, 2009**; **Beetz et al., 2015**; **Xu et al., 2020**). Our model therefore hypothesises that insects possess different compass systems based on varied sensory information and further that insects possess the capability (via CX-based RAs) to coordinate their influence optimally according to the current context. Since the global compass, the local compass and the desired heading of RF share the same visual pathway (OL->AOTU->BU->CX), distinct input and output patterns along this pathway may be found by future neuroanatomical studies. In addition, in the proposed model, the activation of current heading and desired heading of RF overlap in the EB, and therefore separation of activation profiles representing each output (e.g. following methods in **Seelig and Jayaraman, 2015**) presents another meaningful topic for future neurophysiological research.

Closed-loop behavioural studies during which the spatial frequency information of views is altered (similar to **Paulk et al., 2015**) coincident with imaging of key brain areas (**Seelig and Jayaraman, 2013**) offers a means to investigate which neural structures make use of what visual information. Complimentary behavioural experiments could verify the distinct VH and RF systems by selectively blocking the proposed neural pathways with impacts on behaviour predicted by **Figure 2C** and **Figure 4B**, respectively. **Ofstad et al., 2011** report that visual homing abilities are lost for fruit flies with a blocked EB of the CX but not MB, which is predicted by our model if animals have learned target-facing views to which they can later align using their RF guidance system. Analysis of animal's orientation during learning is thus vital to unpacking precisely how the above results arise.

With the elemental guidance strategies defined, we propose that their outputs are coordinated through the combined action of the MBs and CX. Specifically, we demonstrate that a pair of ring attractor networks that have similar connectivity patterns of the CX-based head-direction system

(*Kim et al., 2017*; *Turner-Evans et al., 2019*; *Pisokas et al., 2019*), are sufficient for optimally weighting multiple directional cues from the same frame of reference (e.g. VH and PI). The use of a pair of integrating RAs is inspired by the column structure of the FB which has 16 neural columns divided into two groups of 8 neural columns that each represent the entire 360°space. The optimal integration of PI and VH using a ring attractor closely matches the networks theorised to govern optimal directional integration in mammals (*Jeffery et al., 2016*) and supports hypothesis of their conserved use across animals (*Mangan and Yue, 2018*). Optimality is secured either through adapting the shape of the activity profile of the input as is the case for PI which naturally scales with distance, or by using a standardised input activity profile with cross-inhibition of competing cues as is the case for VH in the model. The later schema avoids the need for ever increasing neural activity to maintain relevance.

To replicate the suite of navigational behaviours described in *Figure 1*, our network includes three independent ring attractor networks: the global compass head direction system *Pisokas et al., 2019*; the local compass head direction system (*Seelig and Jayaraman, 2015*; *Kim et al., 2017*; *Turner-Evans et al., 2019*); and an Off-route integration system (modelled here). We would speculate that it is likely that central place foraging insects also possess a similar integration network for 'On-Route' cues (not modelled here) bringing the total number of RAs to four. The utility of RAs for head-direction tracking arises from their properties in converging activity to a signal bump that can easily be shifted by sensory input and is maintained in the absence of stimulation. In addition, RAs also possess the beneficial property that they spontaneously weight competing sensory information stored as bumps of activity in an optimal manner. Thus, there are excellent computational reasons for insects to invest in such neural structures. Yet, it should be clear that the model proposed here represents a proof-of-concept demonstrating that the underlying network architectures already mapped to the CX (directional cues encoded as bumps of activity *Seelig and Jayaraman, 2015*; *Heinze and Homberg, 2007*; various lateral shifting mechanisms (*Stone et al., 2017*; *Green et al., 2017*; *Turner-Evans et al., 2017*); RAs [*Kim et al., 2017*; *Turner-Evans et al., 2019*; *Pisokas et al., 2019*]) are sufficient to generate adaptive navigation but further studies are required to critique and refine the biological realism of this hypothesis.

While this assemblage recreates optimal integration of strategies that share a compass system, it does not easily extend to integration of directional cues from other frames of reference (e.g. VH and PI reference the global compass versus RF that references a local compass). Indeed as the CX steering network seeks to minimise the difference between a current and a desired heading, calibrating input signals from different frames of reference would require a similar calibration of their respective compass systems. Rather, the proposed model incorporates a context-dependent non-linear switching mechanism driven by the output of the MB that alternates between strategies: global compass based PI and VH are triggered when the surroundings are unfamiliar, but when in familiar surroundings engage local compass-based RF. In summary, the adaptive behaviour demonstrated is the result of distinct guidance systems that converge in the CX, with their relative weighting defined by the output of the MB. This distributed architecture is reminiscent of mechanisms found in the visual learning of honeybees (*Plath et al., 2017*), and supports the hypothesis that the CX is the navigation coordinator of insects (*Heinze, 2017*; *Honkanen et al., 2019*) but shows how the MB acts as a mediator allowing the CX to generate optimal behaviour according to the context.

The resultant unified model of insect navigation *Figure 1B and C* represents a proof-of-principle framework as to how insects might co-ordinate core navigational behaviours (PI, VH and RF) under standard field manipulations *Figure 1A*. Neuroanatomical data has been drawn from across insect classes (see *Table 2*) to ensure neural realism where possible with performance compared to ant navigation behaviour in a single simulated desert ant habitat. The framework can be easily extended to new navigation behaviours observed in other insects from idiothetic PI (*Kim and Dickinson, 2017*) to straight line following *El Jundi et al., 2016* to migrations (*Reppert et al., 2016*) as well as more nuanced strategies that flexibly use directional cues from different sensory modalities (*Wystrach et al., 2013*; *Schwarz et al., 2017*; *Dacke et al., 2019*). A priority of future works should be the investigation of the differences and commonalities in sensory systems, neural structures and ecology of different insect navigators and how they impact behaviour allowing for extension and refinement of the framework for different animals. Complementary stress-testing of models across different environments in both simulation and robotic studies are also required to ensure that model

**Table 2.** The details of the main neurons used in the proposed model.

| Name | Function | Num | Network | Brain region | Neuron in species(e.g.) | Reference |
|---|---|---|---|---|---|---|
| I-TB1 | Global compass current heading | 8 | Ring attractor | | TB1 in Schistocerca gregariaand Megalopta genalis | *Heinze and Homberg, 2008*; *Stone et al., 2017* |
| II-TB1 | Local compass current heading | 8 | Ring attractor | | Δ7 in *Drosophila* | *Franconville et al., 2018* |
| S I-TB1 | Copy of shifted global heading | 8 | Ring | | No data | / |
| VH-L | VH desired heading left | 8 | Ring | | No data | |
| VH-R | VH desired heading right | 8 | Ring | | No data | |
| PI-L | PI desired heading left | 8 | Ring | | CPU4 in Schistocerca gregariaand Megalopta genalis | *Heinze and Homberg, 2008*; *Stone et al., 2017* |
| PI-R | PI desired heading right | 8 | Ring | CX | P-F3N2v in *Drosophila* | *Franconville et al., 2018* |
| RF-L | RF desired heading left | 8 | Ring | | No data | / |
| RF-R | RF desired heading right | 8 | Ring | | No data | |
| RA-L | Cue integration left | 8 | Ring attractor | | No data | |
| RA-R | Cue integration right | 8 | Ring attractor | | No data | |
| CPU1 | Comparing the current and desired heading | 16 | Steering circuit | | CPU1 in Schistocerca gregaria and Megalopta genalis PF-LCre in *Drosophila* | *Heinze and Homberg, 2008*; *Stone et al., 2017* *Franconville et al., 2018* |
| vPN | visual projection | 81 | | | MB neurons in *Drosophila* | *Aso et al., 2014* |
| KCs | Kenyon cells | 4000 | Associative learning | MB | Camponotus | *Ehmer and Gronenberg, 2004* |
| MBON | visual novelty | 1 | | | Apis mellifera | *Rybak and Menzel, 1993* |
| TUN | Tuning weights from PI to RA | 1 | / | | No data | / |
| SN1 | Turn on/off the RF output to CPU1 | 1 | Switch circuit | SMP | No data | |
| SN2 | Turn on/off the RA output to CPU1 | 1 | Switch circuit | | No data | |

performance generalises across species and habitats and to provide guidance to researchers seeking the sensory, processing and learning circuits underpinning these abilities.

## Materials and methods

All source code related to this publication is available for download at https://github.com/Xuelong-Sun/InsectNavigationToolkitModelling (*Sun et al., 2020* ; copy archived at https://github.com/elifesciences-publications/InsectNavigationToolkitModelling). All simulations and network models are implemented by Python 3.5 and make use of external libraries-*numpy*, *matplotlib*, *scipy*, *PIL* and *cv2*.

### Simulated 3D world

The environment used in this study is that provided by *Stone et al., 2018* which is itself adapted from *Baddeley et al., 2012* (see *Figure 7C*). It is a virtual ant-like world consisting of randomly generated bushes, trees and tussocks based on triangular patches (for more details see *Baddeley et al., 2012*). Therefore, the data of this simulated world is stored in a matrix with the size of $N_P \times 3 \times 3$, defining the three dimensional coordinates (x,y,z) of the three vertices of $N_P$ (number of patches) triangle patches. Agent movement was constrained to a $20m \times 20m$ training and test area allowing free movement without the requirement of an additional obstacle avoidance mechanism.

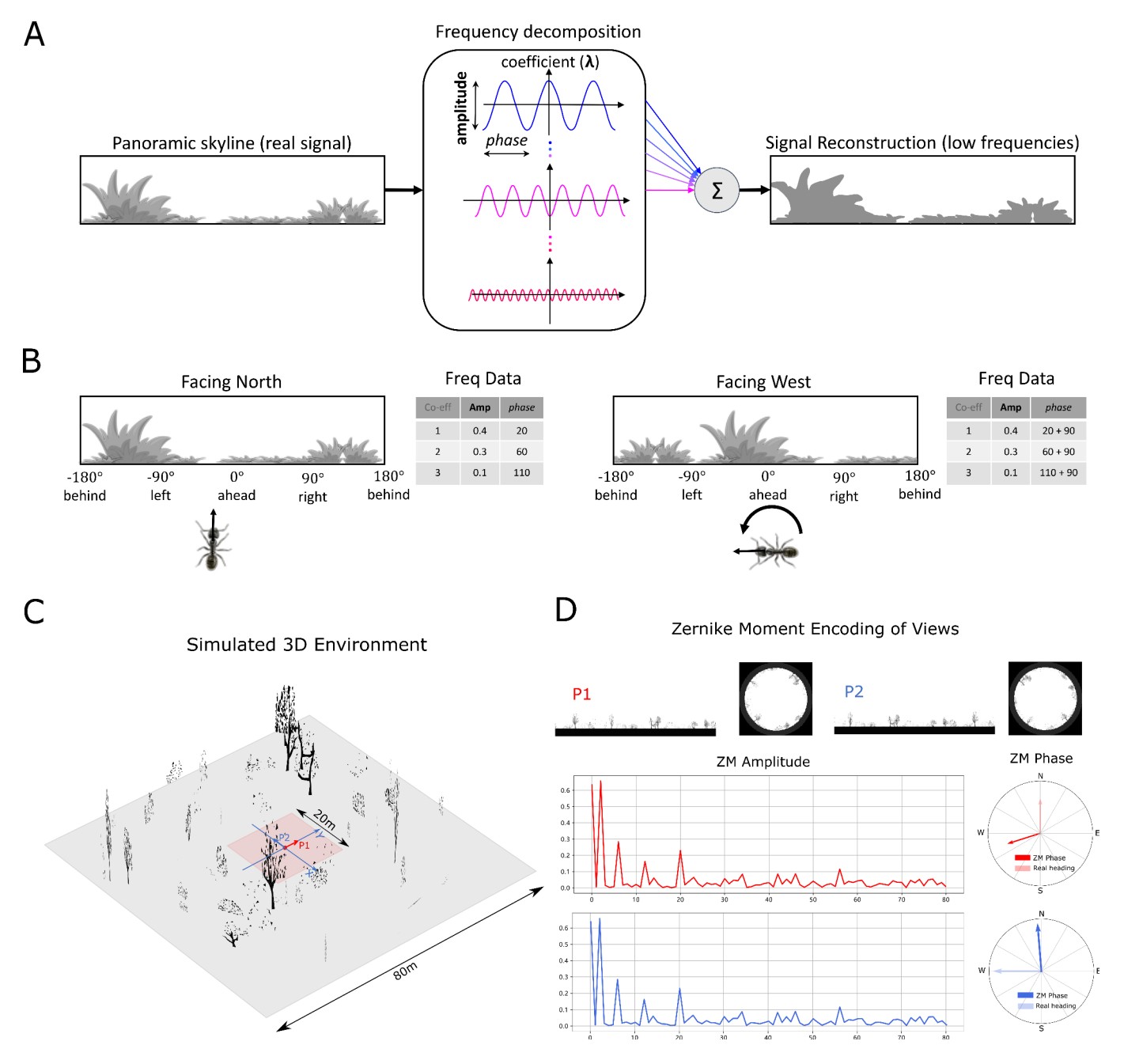

**Figure 7.** Information provided by frequency encoding in cartoon and simulated ant environments. (**A**): A cartoon depiction of a panoramic skyline, it's decomposition into trigonometric functions, and reconstruction through the summation of low frequency coefficients reflecting standard image compression techniques. (**B**): Following a 90° rotation there is no change in the amplitudes of the frequency coefficients but the *phases* of the frequency coefficients track the change in orientation providing a rotational invariant signal useful for visual homing and rotationally-varying signal useful for route following, respectively. (**C**): The simulated 3D world used for all experiments. The pink area (size: $20m \times 20m$) is used for model training and testing zone for models allowing obstacle-free movement. (**D**): The frequency encoding (Zernike Moment's amplitudes and *phase*) of the views sampled from the same location but with different headings (P1 and P2 in (**C**), with 90° heading difference) in the simulated world. The first 81 amplitudes are identical while the *phases* have the difference of about 90°.

The online version of this article includes the following source data for figure 7:

**Source data 1.** The matrix of simulated 3D world.

## Image reconstruction

The agent's visual input at location $(x, y)$ with the heading direction $\theta_h$ is simulated from a point 1 cm above from the ground plane with field of view 360° wide by 90° high (centred on the horizon). This panoramic image ($300 \times 104$) is then wrapped onto a sky-centred disk as required by the Zernike Moments transformation algorithm used with the size of $208(104 \times 2) \times 208$ ready for image processing (see *Figure 7D* upper).

## Image processing

### Frequency encoding conceptual overview

Image compression algorithms such as JPEG encoding *Hudson et al., 2018* have long utilised the fact that a complex signal can be decomposed into a series of trigonometric functions that oscillate at different frequencies. The original signal can then be reconstructed by summing all (for prefect reconstruction) or some (for approximate reconstruction) of the base trigonometric functions. Thus, compression algorithms seek a balance between using the fewest trigonometric functions to encode the scene (for example, by omitting high frequencies that humans struggle to perceive), and the accuracy of the reconstructed signal (often given as an option when converting to JPEG format). *Figure 7A* provides a cartoon of the frequency decomposition process for a panoramic view.

When such transforms are applied to fully panoramic images, or skylines, benefits beyond compression arise. Specifically, discrete transformation algorithms used to extract the frequency information generate a series of information triplets to describe the original function: frequency coefficients describe the frequency of the trigonometric function with associated amplitudes and *phase* values defining the vertical height versus the mean and the lateral position of the waveform respectively (*Figure 7A*). For panoramic views, regardless of the rotational angle of the image capturing device (eye or camera) the entire signal will always be visible and hence the amplitudes of the frequency coefficients do not alter with rotation (*Figure 7B*). This information has been used for successful place recognition in a series of robot studies (*Pajdla and Hlaváč, 1999*; *Menegatti et al., 2004*; *Stone et al., 2016*). Most recently *Stone et al., 2018* demonstrated that the difference between the amplitudes of the frequency coefficients recorded at two locations increases monotonically with distance producing an error surface suitable for visual homing. This feature of the frequency encoding underlies the visual homing results described in Mushroom bodies as drivers of rotational invariant visual homing.

In addition, as the *phase* of each coefficient describes how to align the signal this will naturally track any rotation in the panoramic view (*Figure 7B*) providing a means to realign with previous headings. The *phase* components of panoramic images have been utilised previously to derive the home direction in a visual homing task (*Stürzl and Mallot, 2006*). This feature of the frequency encoding underlies the route following results described in Route following in the insect brain.

The image processing field has created an array of algorithms for deriving the frequency content of continuous signals (*Jiang et al., 1996*; *Gonzalez et al., 2004*). To allow exploration of the usefulness of frequency information, and how it could be used by the known neural structures, we adopt the same Zernike Moment algorithm used by *Stone et al., 2018*, but the reader should be clear that there are many alternate and more biologically plausible processes by which insects could derive similar information. It is beyond the scope of this proof of concept study to define precisely how this process might happen in insects but future research possibilities are outlined in the Discussion.

### Zernike Moments encoding

Zernike Moments (ZM) are defined as the projection of a function onto orthogonal basis polynomials called Zernike polynomials (*Teague, 1980*; *Khotanzad and Hong, 1990*). This set of functions are defined on the unit circle with polar coordinates $(\rho, \theta)$ shown as:

$$V_{nm}(\rho, \theta) = R_{nm}(\rho)e^{jm\theta} \tag{1}$$

where $n \in N^+$ is the order and $m$ is the repetition meeting the condition: $m \in N$, $|m| \leq n$ and $n - |m|$ is even to ensure the rotational invariant property is met. $R_{nm}(\rho)$ is the radial polynomial defined as:

$$R_{nm}(\rho) = \sum_{s=0}^{n-|m|/2} (-1)^s \frac{(n-s)!}{s!(\frac{n+|m|}{2}-s)!(\frac{n-|m|}{2}-s)!} \rho^{n-2s} \tag{2}$$

For a continuous image function $f(x,y)$, the ZM coefficient can be calculated by:

$$Z_{nm}(\rho) = \frac{n+1}{\pi} \int\int_{x^2+y^2 \leq 1} f(x,y) V_{nm}^*(\rho,\theta) dxdy \tag{3}$$

For a digital image, summations can replace the integrals to give the ZM:

$$Z_{nm}(\rho) = \frac{n+1}{\pi} \sum_x \sum_y f(x,y) V_{nm}^*(\rho,\theta), \quad x^2+y^2 \leq 1. \tag{4}$$

ZM are extracted from the simulated insect views in wrapped format (*Figure 7D*) whose centre is taken to be the origin of the polar coordinates such that all valid pixels lie within the unit circle. For a given image $I$ (P1 in *Figure 7D*) and the rotated version of this image $I^\theta$, (P2 in *Figure 7D*), the **amplitude** $A = |Z|$ and *phase* $\Phi = \angle Z$ of ZM coefficients of these two images will satisfy:

$$\begin{cases} |Z_{nm}^{\theta_r}| &= |Z_{nm}e^{-jm\theta_r}| = |Z_{nm}| \quad i.e., \quad A_{nm}^{\theta_r} = A_{nm} \\ \Phi_{nm}^{\theta_r} &= \Phi_{nm} - m\theta_r \end{cases} \tag{5}$$

From which we can see that the **amplitude** of the ZM coefficient remains the same while the *phase* of ZM carries the information regarding the rotation (see *Figure 7A and D*). This property is the cornerstone of the visual navigation model where the **amplitudes** encode the features of the view while the *phase* defines the orientation.

Amplitudes for ZM orders ranging from $n = 0$ to $n = 16$ were selected as they appeared to cover the majority of information within the image. From *Equation 1*, we know that $V_{n,m} = V_{n,-m}$, so we limited $m \in N^+$ to reduce the computational cost, which sets the total number of ZM coefficients ($N_{ZM}$) to $(16 \div 2 + 1)^2 = 81$ which was input to the visual navigation networks. For training the ANN network for RF, in *Equation 5*, if we set $m = 1$, such that $\Phi_{n,1}^{\theta_r} = \Phi_{n,1} - \theta_r$ which means that all ZM coefficients will provide the same information when the image is rotated. Further, the difference between the *phase* of ZM coefficients of the current view with those of the memorised view, will inherently provide the angle with which to turn to realign oneself, that is:

$$\Phi_{7,1}^{current} - \Phi_{7,1}^{memory} = \theta_h - \theta_m \tag{6}$$

where the order $n$ of this ZM is selected to be $n = 7$ manually by comparing the performance with different orders in this specific virtual environment, $\theta_h$ is the current heading of the agent while $\theta_m$ is the memorised heading direction (desired heading direction).

## Neural networks

We use the simple firing rate to model the neurons in the proposed networks, where the output firing rate $C$ is a sigmoid function of the input $I$ if there is no special note. In the following descriptions and formulas, a subscript is used to represent the layers or name of the neuron while the superscript is used to represent the value at a specific time or with a specific index.

### Current headings

In the proposed model, there are two independent compass systems based on the global and the local cues respectively so named global and local compass correspondingly. These two compass systems have similar neural pathways from OL via AOTU and BU to the CX but ended distinct groupings of TB1 neurons: I-TB1 and II-TB1 in the PB.

### Global compass

The global compass neural network applied in this study is the same as that of *Stone et al., 2017*, which has three layers of neurons: TL neurons, CL1 neurons and I-TB1 neurons. The 16 TL neurons respond to simulated polarised light input and are directly modelled as:

$$I_{TL} = \cos(\theta_{TL} - \theta_h) \tag{7}$$

where $\theta_{TL} \in \{0, \pi/4, \pi/2, 3\pi/4, \pi, 5\pi/4, 3\pi/2, 7\pi/4\}$ is the angular preference of the 16 TL-neurons. The 16 CL1-neurons are inhibited by TL-neuron activity which invert the polarisation response:

$$I_{CL1} = 1.0 - C_{TL} \tag{8}$$

The 8 I-TB1 neurons act as a ring attractor creating a sinusoidal encoding of the current heading. Each I-TB1 neuron receives excitation from the CL1 neuron sharing the same directional preference and inhibition from other I-TB1 neurons via mutual connections:

$$W_{I-TB1}^{ij} = \frac{\cos(\theta_{I-TB1}^i - \theta_{I-TB1}^j) - 1}{2} \tag{9}$$

$$I_{I-TB1}^{t,j} = (1-c)C_{CL1}^{t,j} + c\sum_{i=1}^{8} W_{I-TB1}^{ij} C_{I-TB1}^{t-1,j} \tag{10}$$

where $c$ is a balance factor to modify the strength of the inhibition and the CL1 excitation. Finally, the population coding $C_{I-TB1}^{t,j}, j = 0, 1, \ldots 7$ represents the heading of global compass of the agent at time $t$.

## Local compass

The local compass is derived from the terrestrial cues through a similar visual pathway as the global compass and also ends in a ring attractor network. As for the global compass, the local compass heading is directly modelled by the population encoding of II-TB1 neurons:

$$C_{II-TB1}^i = \cos(\Phi_{7,1} - \theta_{II-TB1}^i) \quad i = 0, 1, \ldots 7 \tag{11}$$

where $\theta_{II-TB1}$ is the angular preference of the II-TB1 neurons and $\Phi_{7,1}$ is the *phase* of ZM. Therefore, the firing rate of $C_{II-TB1}$ encodes the heading of the local compass.

### Visual homing

The neural network of visual homing is an associative network constrained by the anatomical structure of the mushroom body (MB) of the insects. In contrast to *Ardin et al., 2016* where a spiking neural network is implemented to model the MB, we apply a simple version of MB where the average firing rates of neurons are used.

The visual projection neurons (vPNs) directly receive the **amplitudes** of the ZM coefficients as their firing rates:

$$C_{vPN}^i = A^i, \quad i = 0, 1, 2 \ldots N_{vPN} \tag{12}$$

where $N_{vPN}$ is the number of the vPN neurons which is the same as the total number of ZM **amplitudes** applied and in this study $N_{vPN} = N_{ZM} = 81$. The $A^i$ denotes the $i^{th}$ **amplitudes** of ZM coefficients.

The vPNs project into Kenyon cells (KC) through randomly generated binary connections $W_{vPN2KC}$, which result in the scenario wherein one KC receives 10 randomly selected vPNs' activation:

$$I_{KC}^j = \sum_{i=0}^{N_{vPN}} W_{vPN2KC}^{ji} C_{vPN}^i \tag{13}$$

where $I_{KC}^j$ denotes the total input current of $j^{th}$ KC from the vPN and the KCs are modelled as binary neurons with the same threshold $Thr_{kc}$:

$$C_{KC} = \begin{cases} 0 & if \ I_{KC} \leq Thr_{KC} \\ 1 & if \ I_{KC} > Thr_{KC} \end{cases} \tag{14}$$

The MBON neuron sums all the activation of Kenyon cells via plastic connections $W_{KC2EN}$:

$$C_{MBON} = \sum_{i=0}^{N_{KC}} W_{KC2MBON}^i C_{KC}^i \tag{15}$$

An anti-Hebbian learning rule is applied for the plasticity of $W_{KC2MBON}$ in a simple way:

$$W_{KC2MBON}^t = W_{KC2MBON}^{t-1} - \eta_{KC2MBON} \quad if \quad C_{KC}^i \geq W_{KC2MBON}^i \tag{16}$$

where $\eta_{KC2MBON}$ is the learning rate. The learning process will happen only when the reward signal is turned on. The activation of EN $C_{MBON}$ represents the familiarity of the current view and the change of the $C_{MBON}$ is defined as:

$$\Delta C_{MBON} = C_{MBON}^t - C_{MBON}^{t-1} \tag{17}$$

$\Delta C_{MBON}$ is used to track the gradient of the familiarity to guide the agent to the more familiar locations by shifting the I-TB1 neurons' activation $C_{I-TB1}$.

$$C_{VH}^i = C_{I-TB1}^j, j = \begin{cases} i + offset & if \quad i + offset \leq 7 \\ i + offset - 7 & otherwise \end{cases} \quad i = 0, 1, ...7 \tag{18}$$

The relationship between the $\Delta C_{MBON}$ and the *offset* is shown as following:

$$offset = \begin{cases} 0 & if \quad \Delta C_{MBON} < 0 \\ \min(\lfloor k_{VH} \Delta C_{MBON} \rfloor, 4) & otherwise \end{cases} \tag{19}$$

## Path integration

The PI model implemented is that published by *Stone et al., 2017*. The core functionality arises from the CPU4 neurons that integrate the activation of TN2 neurons that encode the speed of the agent and the inverted activation of direction-sensitive I-TB1 neurons. The result is that the population of CPU4 neurons iteratively track the distance and orientation to the nest (a home vector) in a format akin to a series of directionally locked odometers.

The firing rate of the CPU4 neurons are updated by:

$$I_{CPU4}^t = I_{CPU4}^{t-1} + r(C_{TN2}^t - C_{I-TB1}^t - k) \tag{20}$$

where the rate of the memory accumulation $r = 0.0025$; the memory loss $k = 0.1$; the initial memory charge of CPU4 neurons $I_{CPU4}^0 = 0.1$.

The input of the TN2 neurons encoding the speed is calculated by:

$$\begin{cases} I_{TN2_L} = [\sin(\theta_h + \theta_{TN2}) \cos(\theta_h + \theta_{TN2})] v \\ I_{TN2_R} = [\sin(\theta_h - \theta_{TN2}) \cos(\theta_h - \theta_{TN2})] v \end{cases} \tag{21}$$

where $v$ is the velocity (see *Equation 39*) of the agent and $\theta_{TN2}$ is the preference angle of the TN2 neurons. In this study $\theta_{TN2} = \pi/4$. The activation function applied to TN2 neurons is the rectified linear function given by:

$$C_{TN2} = \max(0, 2I_{TN2}) \tag{22}$$

As CPU4 neurons integrate the speed and direction of the agent, the desired heading of PI can be represented by the population encoding of these neurons, thus:

$$C_{PI} = C_{CPU4} \tag{23}$$

## Route following

The route following model is based on a simple artificial neural network (ANN) with just one hidden layer. The input layer directly takes the amplitudes of the ZM coefficients as the activation in the same way as that of visual projection neurons in MB network. This is a fully connected neural network with the sigmoid activation function, so the forward propagation is ruled by:

$$\begin{cases} Z_l^i = \sum_{i=0}^{N} W^{ji} Y_{l-1}^j \\ Y_i^l = sigmoid(Z_l^i) = \frac{1}{1+e^{-Z_l^i}} \end{cases} \quad i = 0, 1, ...7 \quad and \quad l = 0, 1, 2 \tag{24}$$

where $Z_l^i$ and $Y_l^i$ denote the input and output of the $i^{th}$ neuron in $l^{th}$ layer, thus the input is the same as the MB network $Z_0^i = A^i, i = 0, 1, ...N_{ZM}$ and the output of the ANN is consequently the population coding of the RF desired heading, that is:

$$C_{RF}^i = Y_i^2 \quad i = 0, 1, ...7 \tag{25}$$

For a fast and efficient implementation, the learning method applied here is back propagation with gradient descend. Training data is derived from the amplitudes and the population encoded *phases* of the ZM coefficients of the images reconstructed along a habitual route. As shown in *Equation 11* the II-TB1 neurons encode the heading of local compass, therefore, the training pair for the RF network can be defined as $\{A, C_{II-TB1}\}$. After training, this network will correlate the desired ZM *phase* with the specific ZM amplitudes, and when RF is running, the output of this neural network $C_{RF}$ will represent the desired heading with respect to the current heading of the local compass represented by the population encoding of II-TB1 neurons.

## Coordination of elemental guidance strategies

The coordination of the three main navigation strategies PI, VH and RF are realised in distinct stages. Firstly, Off-route strategies (PI and VH) are optimally integrated by weighing according to the certainly of each before a context-dependent switch activates either On-route (RF) or Off-route strategies depending on the current visual novelty.

### Optimal cue integration

A ring attractor neural network is used to integrate the cues from the VH and PI guidance systems. As reported in *Hoinville and Wehner, 2018* summation of directional cues represented in vector format leads to optimal angular cue integration which is the same case as real insects. *Mangan and Yue, 2018* gave a biology plausible way to do this kind of computation based on a simple ring attractor neural network. There are two populations of neurons in this network, the first is the integration neurons (IN) which is the output population of the network. Constrained by the number of columns in each hemisphere of the insects CX, we set the number of the IN to be 8, and its firing rate is updated by:

$$\tau \frac{dC_{IN}}{dt} = -C_{IN} + g\left(\sum_{j=1}^{n} W_{E2E}^{ji} C_{IN}^j + X_1^i + X_2^i + W_{I2E} C_{UI}\right) \quad i = 0, 1, ...7. \tag{26}$$

where $W_{E2E}^{ji}$ is the recurrent connections from $j^{th}$ neuron to $i^{th}$ neuron, $g(x)$ is the activation function that provides the non-linear property of the neuron:

$$g(c) = max(0, \rho + c) \tag{27}$$

where $\rho$ denotes the offset of the function.

In *Equation 26*, $X_1$ and $X_2$ generally denote the cues that should be integrated. In this study, $X_1$ and $X_2$ represent the desired heading of path integration ($C_{PI}$) and visual homing ($C_{VH}$). The desired heading of PI is also tuned by the tuning neuron (TUN) in SMP which is stimulated by the MBON of MB (see *Figure 3A*) and its activation function is defined by a rectified linear function, that is:

$$C_{TUN} = min(k_{TUN} C_{EN}, 1) \tag{28}$$

where $k_{TUN}$ is the scaling factor.

Thus, the $X_1$ and $X_2$ for this ring attractor network can be calculated by:

$$\begin{cases} X_1^i = C_{TUN} C_{PI}^i \\ X_2^i = C_{VH}^i \end{cases} \quad i = 0, 1, ...7 \tag{29}$$

The second population of the ring attractor is called the uniform inhibition (UI) neurons modelled by:

$$\tau \frac{dC_{UI}}{dt} = -u + g\left(W_{I2I}C_{UI} + W_{E2I}\sum_{k=1}^{n}C_{IN}^{k}\right) \quad i = 0, 1, ...7.$$

(30)

After arriving at a stable state, the firing rate of the integration neurons in this ring attractor network provides the population encoding of the optimal integrated output $C_{OI}$:

$$C_{OI} = C_{CN}$$

(31)

## Context-dependent switch

The model generates two current/desired headings pairs: the current heading of global compass decoded by $C_{I-TB1}$ with the desired heading optimally integrated by the integration neurons of the ring attractor network $C_{OI}$ and the current heading of local compass decoded by II-TB1 neurons $C_{II-TB2}$ with the desired heading decoded by the output of the RF network $C_{RF}$. These two pairs of signal both are connected to the steering circuit (see *Figure 5A* and Steering circuit) but are turned on/off by two switching neurons (SN1 and SN2) in the SMP (*Figure 5A*). SN2 neuron receives the activation from MBON neuron and is modelled as:

$$SN2 = \begin{cases} 0 & if \quad C_{MBON} < Thr_{SN2} \\ 1 & otherwise \end{cases}$$

(32)

While SN1 will always fire unless SN2 fires:

$$SN1 = \begin{cases} 0 & if \quad C_{SN2} = 1 \\ 1 & otherwise \end{cases}$$

(33)

Therefore, the context-depend switch is achieved according to the current visual novelty represented by the activation of MBON.

## Steering circuit

The steering neurons, that is CPU1 neurons ($C_{CPU1}^{i}, i = 0, 1, 2...15$) receive excitatory input from the desired heading ($C_{DH}^{i}, i = 0, 1, 2...15$) and inhibitory input from the current heading ($C_{CH}, i = 0, 1, 2...15$) to generate the turning signal:

$$C_{ST}^{i} = C_{DH}^{i} - C_{CH}^{i} \quad i = 0, 1, ...15$$

(34)

The turning angle is determined by the difference of the activation summations between left ($i = 0, 1, 2...7$) and right ($i = 8, 9, 10...15$) set of CPU1 neurons:

$$\theta_M = k_{motor}\left(\sum_{i=0}^{7}C_{CPU1} - \sum_{i=8}^{15}C_{CPU1}\right)$$

(35)

which corresponds to the difference of the length of the subtracted left and right vectors in *Figure 2A*. In addition, as it is illustrated in *Figure 2A*, another key part of steering circuit is the left/right shifted desired heading, in this paper, this is achieved by the offset connectivity pattern ($W_{DH2CPU1L}$ and $W_{DH2CPU1R}$) from the desired heading to the steering neurons (*Heinze and Homberg, 2008*; *Stone et al., 2017*):

$$\begin{cases} C_{DH}^{0-7} = C_{SN1}C_{RF}W_{DH2CPU1L} + C_{SN2}C_{OI}W_{DH2CPU1L} \\ C_{DH}^{8-15} = C_{SN1}C_{RF}W_{DH2CPU1R} + C_{SN2}C_{OI}W_{DH2CPU1R} \end{cases}$$

(36)

Where the $W_{DH2CPU1L}$ and $W_{DH2CPU1R}$ are:

$$W_{DH2CPU1L} = \begin{bmatrix} 0 & 1 & 0 & 0 & 0 & 0 & 0 & 0 \\ 0 & 0 & 1 & 0 & 0 & 0 & 0 & 0 \\ 0 & 0 & 0 & 1 & 0 & 0 & 0 & 0 \\ 0 & 0 & 0 & 0 & 1 & 0 & 0 & 0 \\ 0 & 0 & 0 & 0 & 0 & 1 & 0 & 0 \\ 0 & 0 & 0 & 0 & 0 & 0 & 1 & 0 \\ 0 & 0 & 0 & 0 & 0 & 0 & 0 & 1 \\ 1 & 0 & 0 & 0 & 0 & 0 & 0 & 0 \end{bmatrix} \quad W_{DH2CPU1R} = \begin{bmatrix} 0 & 0 & 0 & 0 & 0 & 0 & 0 & 1 \\ 0 & 1 & 0 & 0 & 0 & 0 & 0 & 0 \\ 0 & 0 & 1 & 0 & 0 & 0 & 0 & 0 \\ 0 & 0 & 0 & 1 & 0 & 0 & 0 & 0 \\ 0 & 0 & 0 & 0 & 1 & 0 & 0 & 0 \\ 0 & 0 & 0 & 0 & 0 & 1 & 0 & 0 \\ 0 & 0 & 0 & 0 & 0 & 0 & 1 & 0 \\ 0 & 0 & 0 & 0 & 0 & 0 & 0 & 1 \end{bmatrix} \tag{37}$$

which defines the connection pattern realising the left/right shifting of the desired headings used throughout our model (*Figure 2A*, *Figure 3A*, *Figure 4A*, *Figure 5A* and *Figure 6A*.

The current heading input to the steering circuit is also switched between global and local compass input via the SN1 and SN2 neuron:

$$\begin{cases} C_{CH}^{0-7} = C_{SN1} C_{II-TB1} + C_{SN2} C_{I-TB1} \\ C_{CH}^{8-15} = C_{SN1} C_{II-TB1} + C_{SN2} C_{I-TB1} \end{cases} \tag{38}$$

## Detailed neural connectivity of unified model

*Figure 6A* shows a complete picture of the proposed model. Specifically, it highlights the final coordination system showing that CX computing the optimal navigation output with the modulation from the MB and SMP. In addition, offset connectivity pattern from the desired heading to the steering circuit that underpin the left/right shifting is clearly shown. *Figure 6B and C* shows the network generating the desired heading of RF and VH respectively.

In addition, *Table 2* provides details of all modelled neural circuits with their function and naming conventions with links to biological evidence for these neural circuits where it exists and the animal that they were observed in.

## Simulations

*Equation 35* gives the turning angle of the agent, thus the instantaneous "velocity" ($v$) at every step can be computed by:

$$\boldsymbol{v}^t = S_L [\cos \theta_M^t, \sin \theta_M^t] \tag{39}$$

where $S_L$ is the step length with the unit of centimetres. Note that we haven't defined the time accuracy for every step of the simulations, thus the unit of the velocity in this implementation is $cm/step$ rather than $cm/s$. Then the position of agent $\boldsymbol{P}^{t+1}$ in the Cartesian coordinates for the is updated by:

$$\boldsymbol{P}^{t+1} = \boldsymbol{P}^t + \boldsymbol{v}^t \tag{40}$$

The main parameter settings for all the simulations in this paper can be found in *Table 1*.

### Reproduce visual navigation behaviour

Inspired by the benchmark study of real ants in *Wystrach et al., 2012*, we test our model of VH and RF by reproducing the homing behaviours in that study. This is achieved by constructing a habitual route with a similar shape (arc or banana shape) in our simulated 3D world. The position $\boldsymbol{P}_{R-Arc}$ and heading $\theta_{R-Arc}$ along that route is manually generated by:

$$\begin{cases} \theta_{R-Arc}^i = \frac{\pi}{2} - i \frac{\pi}{2N_M} \\ \boldsymbol{P}_{R-Arc}^i = [-R \sin \theta_{R-Arc}^i, -7 + R \cos \theta_{R-Arc}^i] \end{cases} \quad i = 0, 1 \cdots N_M \tag{41}$$

where the $R = 7m$ is the radius of the arc and $N_M = 20$ in this case is the number of the sampling points where view images are reconstructed along the route. The reconstructed views then be wrapped and decomposed by ZM into amplitudes and *phases* are used to train the ANN network of RF and MB network of VH.

### Visual homing

After training, 12 agents with different initial headings that were evenly distributed in $[0, 360)$ were released at the sideways release point ($\boldsymbol{P} = [0, -7]$) for the simulation of VH (*Figure 2D*). The headings of the agents at radius 2.5 m from the release point (manually selected to ensure that the all the agents have completed any large initial loop) are taken as the initial headings.

### Route following

After training, 2 agents with 0° and 180° are released at the different release points ($\boldsymbol{P} = [-9, -7], [-8, -7], [-7, -7], [-6, -7], [-5, -7]$) for the simulation of RF (see *Figure 4B*) to generate the homing path. And then, we release 12 agents on the route ($\boldsymbol{P} = [-7, -7]$) with different initial headings that is evenly distributed in $[0, 360)$ to compare the results with the real ant data in *Wystrach et al., 2012*. The heading of each agent at the position that is 0.6m from the release point is taken as the initial heading.

### Reproduce the optimal cue integration behaviour

We evaluated the cue integration model by reproducing the results of *Wystrach et al., 2012* and *Legge et al., 2014*. The ants' outbound routes in *Wystrach et al., 2015* is bounded by the corridor, so here we simulate the velocity of the agent by:

$$\boldsymbol{v}_{out}^t = [rand(0, 2V_0) - V_0, V_0], \quad t = 0, 1 \ldots T_{out} \tag{42}$$

where the function $rand(0, x)$ generates a random value from the uniform distribution of $[0, x]$, thus the speed of x-axis will be in $[-V_0, V_0]$ and will cancel each other during the forging. The speed of y-axis is constant so it will accumulated and be recorded by the PI model. And $V_0 = 1cm/step$ is the basic speed of the agent and $T_{out}$ is the total time for outbound phase determining the length of the outbound route. As for the simulated homing route, we duplicate the outbound route when $T_{out} = 300$ but with a inverted heading direction. And then the visual navigation network was trained with images sampled along a simulated route (grey curve in *Figure 3B*).

### Tuning PI uncertainty

The agent in this simulation was allowed to forage to different distances of 0.1m, 1m, 3m or 7m from the nest to accrue different PI states and directional certainties before being translated to a never-before-experienced test site 1.5m from the nest. (RP1 in *Figure 3B*). For each trial, we release 20 agents with different initial headings that is evenly distributed in $[0, 360)$. The headings of every agent at the position that is 0.6m from the start point is taken as the initial headings, and the mean direction and the 95% confidential intervals are calculated. As in the biological experiment, the angle between the directions recommended by the PI and visual navigation systems differed by approximately 130°.

As the length of the home vector increase (0.1m -> 7m) the activation of PI memory becomes higher (*Figure 3B*), and increasingly determines the output of the ring attractor integration. Since the length of the home vector is also encoded in the activation of the PI memory neurons, the ring attractor can extract this information as the strength of the cue. As the visual familiarity is nearly the same in the vicinity of the release point, the strength of visual homing circuit remains constant and has more of an influence as the PI length drops.

### Tuning visual uncertainty

The agent in this simulation was allowed to forage up to 1m from the nest to accrue its PI state and directional certainty before being translated to three different release points (RP1, RP2 and RP3 in *Figure 3B*). As the distance from nest increases (RP1->RP2->RP3) so does the visual uncertainty. For each trial, we release 12 agents with different initial headings that is evenly distributed in $[0, 360)$. The headings of each agent at the position that is 0.3m from the start point is taken as the initial headings, and the mean direction and the 95% confidential intervals are calculated.

### Whole model

The simulated habitual route remains the same as in the simulation of visual navigation (Reproduce visual navigation behaviour) as is the learning procedure. The zero- and full- vector agents are both released at $[-2, -7]$ with the heading 0° and 90°, respectively. The full-vector agent's PI memory is generated by letting the agent forage along the route from nest to feeder.

## Acknowledgements

This research has received funding from the European Union's Horizon 2020 research and innovation programme under the Marie Sklodowska-Curie grant agreement No 778062, ULTRACEPT and No 691154, STEP2DYNA.

Thanks to Barbara Webb and Insects Robotics Group at the Univ of Edinburgh, Hadi Maboudi, Alex Cope and Andrew Philippedes for comments on early drafts, and to Antoine Wystrach for provision of data from previous works. Thanks for proof readers Anne and Mike Mangan (Snr). Finally, thanks to our editor and reviewers who helped improve the model and manuscript through their excellent feedback.

## Additional information

### Funding

| Funder | Grant reference number | Author |
| --- | --- | --- |
| Horizon 2020 Framework Programme | ULTRACEPT 778062 | Xuelong Sun Shigang Yue |
| Horizon 2020 Framework Programme | STEP2DYNA 691154 | Xuelong Sun Shigang Yue |

The funders had no role in study design, data collection and interpretation, or the decision to submit the work for publication.

### Author contributions

Xuelong Sun, Conceptualization, Data curation, Software, Formal analysis, Validation, Investigation, Visualization, Methodology, Writing - original draft, Writing - review and editing; Shigang Yue, Supervision, Funding acquisition, Project administration, Writing - review and editing; Michael Mangan, Conceptualization, Resources, Data curation, Supervision, Validation, Investigation, Visualization, Methodology, Writing - original draft, Writing - review and editing

### Author ORCIDs

Xuelong Sun https://orcid.org/0000-0001-9035-5523

### Decision letter and Author response

Decision letter https://doi.org/10.7554/eLife.54026.sa1
Author response https://doi.org/10.7554/eLife.54026.sa2

## Additional files

### Supplementary files

• Source code 1. Data_Model_Simulations_Results.

• Transparent reporting form

### Data availability

All the source code of the implementation and part of the data are uploaded to Github and are available via https://github.com/XuelongSun/InsectNavigationToolkitModelling (copy archived at https://github.com/elifesciences-publications/InsectNavigationToolkitModelling).

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
