## [Decision Letter]

**Acceptance summary:**

This work builds upon prior models to propose an integrated model for computational strategies used by insects to perform their remarkable navigational feats. This new model accounts several capabilities and the flexibilities that accomplished insect navigators display such as visual homing, which were not as well accounted for earlier. The integrated model is not only an important addition to the literature on the insect central complex, but also particularly valuable because it pins specific computational functions on specific anatomical structures, making predictions that are potentially testable in the near-medium term.

**Decision letter after peer review:**

Thank you for submitting your article "A Decentralised Neural Model Explaining Optimal Integration Of Navigational Strategies in Insects" for consideration by *eLife*. Your article has been reviewed by three peer reviewers, and the evaluation has been overseen by Mani Ramaswami as Reviewing Editor and Michael Eisen as the Senior Editor. The following individual involved in review of your submission has agreed to reveal their identity: Stanley Heinze (Reviewer #2).

The reviewers have discussed the reviews with one another and the Reviewing Editor has drafted this decision to help you prepare a revised submission.

Summary:

This is an original, focussed and timely study on a topic of considerable interest: computational strategies used by insects to perform their remarkable navigational feats. The authors identify shortcomings in existing models -specifically, that they do not account for the entire range of capabilities and the flexibility that the most accomplished of insect navigators display such as visual homing, i.e., the ability of the ant to return to familiar region from novel locations. They then integrate and build upon prior models to successfully fill these gaps. The integrated model is particularly valuable because it pins specific computational functions on specific anatomical structures, most notably the central complex and the mushroom body. It is an important addition to both the literature on the insect central complex, as well as to more theoretical navigational work, in particular as many predictions can be made based on the presented models making it, in principle, testable in the near-medium term. The figures are well made and the writing is compact. Nevertheless, several points need to be addressed before publication.

Essential revisions:

1) Accessibility to a broad readership. While the general text is written very well and the content is highly interesting for a life science (in particular insect neuroscience) audience, the Materials and methods section and some aspects of the reasoning behind the model are very technical. Even insect neurobiologists among the reviewers struggled to follow large parts of the methods and had never heard of Zernike moments for instance. The text should be revised to include some more intuitive and broadly accessible language that would allow a biologist to grasp at least the key principles of what is done by those initial analyses of the visual information in the model. A schematic illustration as to what Zernike moments are, maybe combined with some simple examples might help a lot. This is important as the paper is not only directed towards computational biologists, but is highly relevant also for physiologists, anatomists and behavioralists, most of whom would probably fail to grasp the essence of the new principles presented. In similar vein, the authors should ask a mammalian researcher to read the article and provide them with feedback on how accessible they found it. Simple terminology/concepts/structure names in the Abstract/Introduction should not be used until they have been introduced properly, e.g., 'route following', 'visual homing', 'anterior optic tubercle'.

2) On a similar note, the article builds on a lot of prior modeling literature in the insect navigation field, particularly the work from Barbara Webb and colleagues. Important concepts/algorithmic strategies need to be more fully explained here (with appropriate citations) rather than just being referred to in the prior literature. The Materials and methods section does a good job of this, but the Results section could benefit from more explanation to guide unfamiliar readers.

3) It is entirely reasonable that the authors combine experimental and modeling work from a range of different insect species to build different pieces of their own model. By and large they are careful to state which is which. However, they could make it clearer which assumptions are based on experimental data and which are based on prior models (i.e., not actual data). As an example, although the mushroom body has been suggested by numerous modeling studies and conceptually driven reviews to be involved in visual navigation, the experimental evidence for this is lacking, and their precise role is far from well-established.

4) It is excellent that the authors integrate useful components from prior models to construct their integrated model. Although the figures go some way towards clarifying how the different pieces might fit together, it would be useful to make even clearer what is entirely novel here and what is derived/integrated from previous work. In addition, although the authors make a testable case for the involvement of the fan-shaped body in a series of different navigational computations, controlled by the mushroom body, the figures are still somewhat complex and confusing. These should be clarified for the broader readership.

5) Neuroanatomical correspondence of model details: The paper claims that the model is in most parts biologically constrained and that most elements can be mapped onto known neurons. Where this was not possible (route following) the authors speculated about the possible implementations. While on the levels of neuropil groups this is all quite true, the details, especially in the central complex, are less clear and many of the proposed circuits have no known counterpart in any insect brain to date. This is not saying that those parts of the model are not realistic or interesting, but that the claim that they correspond to existing neurons in the central complex, is slightly misleading. Below series of obvious mix ups of cell types below, which need to be corrected (5.1), but additionally, it should be clearly stated where the model does not (yet) have a solid grounding in biology (see point 5.2). Finally, the speculative route following implementation seems at odds with neurophysiological data from various species and alternative pathways and implementations seem more likely (point 5.3).

5.1) Subsection “Mushroom Bodies As Drivers of Rotational Invariant Visual Homing”: CPU3 neurons are supposed to be a mirrored TB1 ring attractor network? Is this really what the authors want to say? CPU3 neurons are known in locusts (Heinze and Homberg, 2008), but connect the PB with the FB as columnar cells. If the authors mean CPU4 cells, these neurons are also not forming a ring-network (even though they could receive shifted compass information from TB1 cells by some means). Most simply, would not a parallel set of TB1 cells be optimally suited for this task? There are four TB1 cells for each column in the PB, potentially enough for four parallel ring attractors. These cells are neurochemically distinct and could function independently (see Beetz et al., 2015).

– There is no known direct connection between the EB and the FB (proposed in Figure 4).

– There is no direct connection from the OL to the CX (indicated in the legend of Figure 1 as underlying PI).

– Subsection “Celestial current heading”: CL2 neurons should be CL1 (CL2 correspond to fly P-EN neurons, not E-PG)

– In the PI section of the Materials and methods, sometimes TN cells are referred to as TN2 cells or just as TN cells. TN2 is one of two types of TN cells (tangential noduli neurons) and was the one primarily used for the standard model of Stone et al., 2017. Please be consistent. Also, the tuning cells of the visual homing circuit are called TN cells. This is very confusing and should be changed.

5.2) There are no known ring attractors in the FB. The only ring attractor shown experimentally is the one in the EB/PB, which employs recurrent feedback loops with the PB (E-PG/P-EN/P-EG cells; equal to CL1a, CL2, and CL1b) and inhibitory neurons in the PB (TB1 or delta7 cells). While a similar recurrent connection pattern is thinkable in the FB as well, using unknown types of columnar cells, there is no experimental support for that. Pontine cells might also form local connections that could result in a RA, but that is even more speculative. Please clearly state that the numerous RAs required by the model are hypothetical and have not yet any biological correspondence in the form of identified cell types. Also, I suppose not all the neuron rings drawn in the figures are ring attractors. I suggest to make that distinction more clear (the many abbreviations for the different neuron rings do not make this easier to follow either).

5.3) The authors assume a second compass system in the PB that is fed directly from the OL via the posterior optical tract. There is no evidence for this beyond a single cell type from locusts that connects the accessory medulla (circadian clock) to the POTU, which is also innervated by TB1 neurons. However, there is no connection to the visual part of the OL, and no physiological data exists on the AME->POTU connection. In contrast, the anterior optic tract via the AOTU has been shown in *Drosophila* to contain many neurons that respond to visual features and they converge on the head direction cells in the EB via a recently resolved mechanism. It seems odd to ignore this known compass pathway and propose another one for which no evidence exists. That said, the authors use the anterior pathway to construct a desired heading via an ANN residing in the AOTU/BU pathway, information that is then used to feed into an EB ring attractor that then connects to additional attractors in the FB. Whereas the EB attractor (in conjunction with the PB) exists, there is no evidence for FB based ring attractors and there is no known direct connection between the EB and the FB. While this all results in a really nice figure, it unfortunately is misleading and based on not enough evidence to show it so prominently (readers might easily take it for factual).

It is useful to point out that there is an alternative solution for at least the compass problem: There are four individual CL1 cells in each column of the EB in locusts as well as in flies (EPG/PEG cells). While they are identical in their projection patterns, some connect the PB to the EB and others connect the EB to the PB, so that there are in theory enough cells to form two parallel recurrent loops (needed to maintain a head direction signal). One of them could be driven by landmarks, while the other could be driven by global compass cues. Whereas the current idea is that both inputs converge on a single head direction signal (celestial and local cue based), this might not be true, given that local cues have been tested in *Drosophila* and global cues in locusts and some other species. These neurons are neurochemically distinct and most likely play different functional roles.

Finally with respect to the desired heading, a short term plasticity based, associative mechanism linking the phase of the head direction signal and the local environment was recently demonstrated in *Drosophila* (Fisher at al., 2019 and Kim et al., 2019). The authors state that several of these phases can be stored and retrieved in each respective environment. This sounds very close to what the authors of the current study suggest for routes in ants. The authors should consider these points and revise the proposed circuit identity accordingly.

6) The overall layout of the model could be further clarified. The authors present many (nicely illustrated) parts of the model, but it is difficult to reconcile some of the partial models with one another and there is no immediate way of seeing how many neurons there are overall, or what their complete connectivity patterns might be. This may be obvious from the code itself, but behavioural biologists, neuroanatomists and physiologists need to be provided more direct intuition for the circuits. The absence of this information hinders independent interpretation and finding alternative solutions for mapping the model onto anatomical neural circuits once newly discovered neurons become available in the future. One possibility is to include (at least in the supplements) a full graphical depiction of the model with all existing neurons and their connections. Maybe using a force directed graph diagram like used by the authors of Stone et al., 2017 for their path integration model results in a model illustration that is intuitively understandable for researchers who think more in terms of anatomy. But even if it turns out to be somewhat messy, it would still be helpful.

7) The authors' could derive more constraints from the fly physiology literature than they do. As examples, Fisher et al., 2019 and Kim et al., 2019 have relevant findings relating to plasticity in mapping visual stimuli onto a compass representation. Turner-Evans et al., 2017 has a data-driven ring attractor model that is relevant, and Turner-Evans, 2019 features data demonstrating that the fly compass for current heading relies on visual input from the anterior optic tubercle, contrary to the authors' assumption deriving from an anatomical pathway from the posterior optic tubercle to the protocerebral bridge (175-176). On a somewhat related note, the fly heading system does not necessarily show 'bar following' in open loop: the experiments cited (Seelig and Jayaraman, 2015) were performed in closed loop, with the animal controlling bar position.

8) The authors should also release and include an properly commented code that they used for modelling in the final submission.

9) Why is the velocity of the simulated ant (Vo = 1cm/s) so much slower than that of the real one (about 50cm/s)? This point must be discussed. Is there any fundamental reason?

10) What would happen to the simulated ant if an obstacle was placed on the familiar route ? What is the robustness of the Zernike-based moment algorithm to the unpredicted presence of an obstacle that could appear during the homing ? Additional simulations to address this issue could show the robustness of the proposed navigation model. These new simulations could be in line with the well-known experiments proposed by Wehner and Wehner (R. Wehner and S. Wehner, “Insect navigation : use of maps or Ariadne's thread?”).

11) Subsection “ANN network and Route Following”: would it be possible to plot Crf with respect to angular orientation of the simulated ant in various place (every 10° steps for example).

[Editors' note: further revisions were suggested prior to acceptance, as described below.]

Thank you for submitting your article "A Decentralised Neural Model Explaining Optimal Integration of Navigational Strategies in Insects" for consideration by *eLife*. Your article has been reviewed by two peer reviewers, and the evaluation has been overseen by Mani Ramaswami as Reviewing Editor and Michael Eisen as the Senior Editor. The following individuals involved in review of your submission have agreed to reveal their identity: Stanley Heinze (Reviewer #2).

The reviewers have discussed the reviews with one another, expressed enthusiasm and appreciation for the revisions you made to the original submission, but still have some suggestions you should consider for further improvement of this article. The Reviewing Editor has drafted this decision to help you prepare a revised submission.

Summary:

This is an original, focussed and timely study on a topic of considerable interest: computational strategies used by insects to perform their remarkable navigational feats. The authors identify shortcomings in existing models -specifically, that they do not account for the entire range of capabilities and the flexibility that the most accomplished of insect navigators display such as visual homing, i.e., the ability of the ant to return to familiar region from novel locations. They then integrate and build upon prior models to successfully fill these gaps. The integrated model is particularly valuable because it pins specific computational functions on specific anatomical structures, most notably the central complex and the mushroom body. It is an important addition to both the literature on the insect central complex, as well as to more theoretical navigational work, in particular as many predictions can be made based on the presented models making it, in principle, testable in the near-medium term. The figures are well made and the writing is compact, the revisions are extensively, carefully done and, after minor edits, the paper should be ready for publication.

Larger points for consideration (Optional revisions at the authors’ discretion):

1) For increased accessibility and readability, please consider doing a little more with the earliest schematics to properly orient the broader readership. As an example, Figure 2A is fairly complicated for a reader who isn't familiar with the insect brain, and the panels at right will not be easy for most people to digest without Stone et al., 2017, open nearby (although the Stone et al. study is a must-read for anyone interested in insect navigation, this may not be the ideal way to get people to read the paper!). Could 'Shifted I-TB1' be unpacked a little by showing how the anatomy of the PB-FB columnar neurons might naturally facilitate the shift (as highlighted in the Stone et al. paper)-perhaps this could be a little breakout box at right. Note that the TB neurons should, in any case, be shown in the PB not the FB.

2).Similarly, although the vector subtraction plots below are helpful to the informed reader, it is not clear that they would be sufficiently explanatory to a newer reader. Again, it is the authors' decision whether or not to do more here.

3) A bigger, conceptual point: it is not obvious that one needs as many additional near-independent ring attractors as are invoked in this model, leaving aside the issue that they seem unlikely from an anatomical perspective. It is not clear or convincing that multiple ring attractors are needed to implement the authors' ideas. This potential opportunity for parsimony deserves some exploration, but the authors can decide whether that's something they want to do as part of this paper or not. The one thing that would be good is to make clear where the additional ring attractors reside in the authors' model: if they are speculatively placed in the FB, that should be made clearer in the early schematics (and also made clear that it is speculation at this stage).

---

## [Author Response]

Essential revisions:1) Accessibility to a broad readership. While the general text is written very well and the content is highly interesting for a life science (in particular insect neuroscience) audience, the Materials and methods section and some aspects of the reasoning behind the model are very technical. Even insect neurobiologists among the reviewers struggled to follow large parts of the Materials and methods and had never heard of Zernike moments for instance. The text should be revised to include some more intuitive and broadly accessible language that would allow a biologist to grasp at least the key principles of what is done by those initial analyses of the visual information in the model. A schematic illustration as to what Zernike moments are, maybe combined with some simple examples might help a lot. This is important as the paper is not only directed towards computational biologists, but is highly relevant also for physiologists, anatomists and behavioralists, most of whom would probably fail to grasp the essence of the new principles presented.

We have added a smoother introduction to results arising from frequency encoded views: specifically in the sections titled Visual Homing and Route Following. In addition, we have added a completely new section to the Materials and methods titled "Frequency Encoding Conceptual Overview" which provides an intuitive description of frequency encoding that prefaces the mathematical description of the Zernike Moment method. This section is now accompanied by an updated Figure 7 which includes both a cartoon depicting frequency encoding followed by the real Zernike Moments encoded of skylines from the simulated world. Finally, we have added a paragraph to the Discussion which looks at how the model could be improved which includes a discussion of more biologically plausible methods for frequency encoding. We believe that this part of our contribution should be much clearer to the non-expert reader now.

In similar vein, the authors should ask a mammalian researcher to read the article and provide them with feedback on how accessible they found it.

We have discussed the presentation of our data to other researchers both with experience in insect and mammalian navigation both personally and via presenting (e.g. student presentation at the recent NeuroMatch conference). These discussions have helped us immensely revise our presentation of challenging concepts – see new introduction to frequency encoding (section "Frequency Encoding Conceptual Overview") as an example.

Simple terminology/concepts/structure names in the Abstract/Introduction should not be used until they have been introduced properly, e.g., 'route following', 'visual homing', 'anterior optic tubercle'.

The Abstract has been updated with definitions of route-following and visual homing and we have made it clearer that we map these behaviours to specific neural circuits in the insect brains e.g. Mushroom Bodies and Anterior Optic Tubercle. Upon re-reading the text we believe that this issue largely arose from pointing readers to Figure 1 before definitions were made in the Results. Hence, we have updated Figure 1 and Figure 1’s legend to provide a more intuitive introduction to our model and associated brain areas with more clearer labelling and definitions. Similar edits have been made in the Introduction which we believe when taken together addresses this issue.

2) On a similar note, the article builds on a lot of prior modeling literature in the insect navigation field, particularly the work from Barbara Webb and colleagues. Important concepts/algorithmic strategies need to be more fully explained here (with appropriate citations) rather than just being referred to in the prior literature. The Materials and methods section does a good job of this, but the Results section could benefit from more explanation to guide unfamiliar readers.

As suggested we have added a detailed description of the functioning of the steering circuit as outlined by Stone et al., 2017, where it is first used (see section titled "Mushroom Bodies as Drivers of Rotational Invariant Visual Homing"). This is accompanied by an additional panel in Figure 2 depicting the steering circuit function in vector format. We have also described a more detailed explanation of the neurophysiological and functional advances made in Stone et al., 2017, regarding the function of PI (see section titled: "Optimally Integrating Visual Homing and Path Integration").

3) It is entirely reasonable that the authors combine experimental and modeling work from a range of different insect species to build different pieces of their own model. By and large they are careful to state which is which. However, they could make it clearer which assumptions are based on experimental data and which are based on prior models (i.e., not actual data). As an example, although the mushroom body has been suggested by numerous modeling studies and conceptually driven reviews to be involved in visual navigation, the experimental evidence for this is lacking, and their precise role is far from well-established.

We have added clarification to the text describing our MB model which we believe is the only section based on only prior models. Further we have added clarification of what neural paths are known and those that we speculate through the use of dashed (speculated) and solid (known) connections throughout our figures. In addition, we have added Table 2 which details the neurophysiological studies on which we base our models making it clear which elements are biologically known and those that are hypothesised.

4) It is excellent that the authors integrate useful components from prior models to construct their integrated model. Although the figures go some way towards clarifying how the different pieces might fit together, it would be useful to make even clearer what is entirely novel here and what is derived/integrated from previous work.

As part of our update to all figures we introduced a star labelling of circuits to indicate which elements were derived from previous works, which were completely novel, and those that are a mixture e.g. previous circuit but used in a novel way, adapted, integrated with other systems. See Figure 2 for its first usage.

In addition, although the authors make a testable case for the involvement of the fan-shaped body in a series of different navigational computations, controlled by the mushroom body, the figures are still somewhat complex and confusing. These should be clarified for the broader readership.

Each of the main figures and their legends have been revised and we now believe that they should be much clearer now. In addition, the new added Figure 6 should be helpful to show the neural connections in fan-shape body.

5) Neuroanatomical correspondence of model details: The paper claims that the model is in most parts biologically constrained and that most elements can be mapped onto known neurons. Where this was not possible (route following) the authors speculated about the possible implementations. While on the levels of neuropil groups this is all quite true, the details, especially in the central complex, are less clear and many of the proposed circuits have no known counterpart in any insect brain to date. This is not saying that those parts of the model are not realistic or interesting, but that the claim that they correspond to existing neurons in the central complex, is slightly misleading. Below series of obvious mix ups of cell types below, which need to be corrected (5.1), but additionally, it should be clearly stated where the model does not (yet) have a solid grounding in biology (see point 5.2). Finally, the speculative route following implementation seems at odds with neurophysiological data from various species and alternative pathways and implementations seem more likely (point 5.3).

This feedbacks is very helpful, thanks very much. See below the changes made accordingly.

5.1) Subsection “Mushroom Bodies As Drivers of Rotational Invariant Visual Homing”: CPU3 neurons are supposed to be a mirrored TB1 ring attractor network? Is this really what the authors want to say? CPU3 neurons are known in locusts (Heinze and Homberg, 2008), but connect the PB with the FB as columnar cells. If the authors mean CPU4 cells, these neurons are also not forming a ring-network (even though they could receive shifted compass information from TB1 cells by some means). Most simply, would not a parallel set of TB1 cells be optimally suited for this task? There are four TB1 cells for each column in the PB, potentially enough for four parallel ring attractors. These cells are neurochemically distinct and could function independently (see Beetz et al., 2015).

Thanks for the feedback. We have changed the text (subsection “Mushroom bodies as drivers of rotational invariant visual homing”, fourth paragraph). See also the response to point 5.3.

– There is no known direct connection between the EB and the FB (proposed in Figure 4)

We have amended both the text and figure to show that this connection is included in our hypothesised pathways but also cite Hanesch et al., 1989, who show evidence for such a connection.

– There is no direct connection from the OL to the CX (indicated in the legend of Figure 1 as underlying PI).

We have amended the model pathway accordingly to 'OL->AUTO->LAL->CX'.

– Subsection “Celestial current heading”: CL2 neurons should be CL1 (CL2 correspond to fly P-EN neurons, not E-PG)

Changed labels to 'CL1' as suggested

– In the PI section of the Materials and methods, sometimes TN cells are referred to as TN2 cells or just as TN cells. TN2 is one of two types of TN cells (tangential noduli neurons) and was the one primarily used for the standard model of Stone et al., 2017. Please be consistent. Also, the tuning cells of the visual homing circuit are called TN cells. This is very confusing and should be changed.

We have now changed all TN to be TN2.

5.2) There are no known ring attractors in the FB. The only ring attractor shown experimentally is the one in the EB/PB, which employs recurrent feedback loops with the PB (E-PG/P-EN/P-EG cells; equal to CL1a, CL2, and CL1b) and inhibitory neurons in the PB (TB1 or delta7 cells). While a similar recurrent connection pattern is thinkable in the FB as well, using unknown types of columnar cells, there is no experimental support for that. Pontine cells might also form local connections that could result in a RA, but that is even more speculative. Please clearly state that the numerous RAs required by the model are hypothetical and have not yet any biological correspondence in the form of identified cell types. Also, I suppose not all the neuron rings drawn in the figures are ring attractors. I suggest to make that distinction more clear (the many abbreviations for the different neuron rings do not make this easier to follow either).

Thanks for this feedback. We only propose one new ring attractor in our model which is used to combine PI and VH signals, but on re-reading our text we see that this was not clear. We have added a sentence to the manuscript where you suggested to clarify the difference between ring networks and ring attractor networks. In addition, we have added labels to the figures to clearly indicate where ring attractors are used. Finally, the new Table 2 also provides of description of each circuit element e.g. ring network vs. ring attractor network, and also their biological supports.

5.3) The authors assume a second compass system in the PB that is fed directly from the OL via the posterior optical tract. There is no evidence for this beyond a single cell type from locusts that connects the accessory medulla (circadian clock) to the POTU, which is also innervated by TB1 neurons. However, there is no connection to the visual part of the OL, and no physiological data exists on the AME->POTU connection. In contrast, the anterior optic tract via the AOTU has been shown in *Drosophila* to contain many neurons that respond to visual features and they converge on the head direction cells in the EB via a recently resolved mechanism. It seems odd to ignore this known compass pathway and propose another one for which no evidence exists. That said, the authors use the anterior pathway to construct a desired heading via an ANN residing in the AOTU/BU pathway, information that is then used to feed into an EB ring attractor that then connects to additional attractors in the FB. Whereas the EB attractor (in conjunction with the PB) exists, there is no evidence for FB based ring attractors and there is no known direct connection between the EB and the FB. While this all results in a really nice figure, it unfortunately is misleading and based on not enough evidence to show it so prominently (readers might easily take it for factual).It is useful to point out that there is an alternative solution for at least the compass problem: There are four individual CL1 cells in each column of the EB in locusts as well as in flies (EPG/PEG cells). While they are identical in their projection patterns, some connect the PB to the EB and others connect the EB to the PB, so that there are in theory enough cells to form two parallel recurrent loops (needed to maintain a head direction signal). One of them could be driven by landmarks, while the other could be driven by global compass cues. Whereas the current idea is that both inputs converge on a single head direction signal (celestial and local cue based), this might not be true, given that local cues have been tested in *Drosophila* and global cues in locusts and some other species. These neurons are neurochemically distinct and most likely play different functional roles.

This is very interesting and helpful price of feedback. Thank you. We have amended our model to generate the terrestrial heading (local compass) pathway to OL->AOTU->BULB->EB->PB to be consistent with the neural data. We have updated our model accordingly which can be seen in Figure 1C, Figure 4A also Figure 6A. Specifically there are four individual TB1/Δ7 cells in each column, which can be used to represent different current-headings. We apply one pathway to generate the Global Compass (celestial heading for VH and PI, I-TB1 neurons), another to generate Local Compass (terrestrial heading for RF, II-TB1 neurons). We have also update the text in the description of the models and in the Discussion.

Finally with respect to the desired heading, a short term plasticity based, associative mechanism linking the phase of the head direction signal and the local environment was recently demonstrated in *Drosophila* (Fisher et al., 2019 and Kim et al., 2019). The authors state that several of these phases can be stored and retrieved in each respective environment. This sounds very close to what the authors of the current study suggest for routes in ants. The authors should consider these points and revise the proposed circuit identity accordingly.

Thank you for this feedback. We have now added these references to our discussion of the possible biological evidence for, and biological pathways for this type of local compass information (see section "Route Following in the Insect Brain"), and adapted our model accordingly. These works have helped us formulate discussions of insects possessing multiple compass systems and the insects’ ability to correlate the views with the orientations, which was very helpful. Finally, we have added some text to the Discussion inspired by these works regarding identifying which groups of neurons in the CX might be encoding for current vs. desired headings.

6) The overall layout of the model could be further clarified. The authors present many (nicely illustrated) parts of the model, but it is difficult to reconcile some of the partial models with one another and there is no immediate way of seeing how many neurons there are overall, or what their complete connectivity patterns might be. This may be obvious from the code itself, but behavioural biologists, neuroanatomists and physiologists need to be provided more direct intuition for the circuits. The absence of this information hinders independent interpretation and finding alternative solutions for mapping the model onto anatomical neural circuits once newly discovered neurons become available in the future. One possibility is to include (at least in the supplements) a full graphical depiction of the model with all existing neurons and their connections. Maybe using a force directed graph diagram like used by the authors of Stone et al., 2017, for their path integration model results in a model illustration that is intuitively understandable for researchers who think more in terms of anatomy. But even if it turns out to be somewhat messy, it would still be helpful.

That’s a really good suggestion. Thanks. We have added both a new force-directed-graph (Figure 6, Materials and methods section) and a table (Table 2, Materials and methods section) that clarifies the type and function of all neurons and their connection that comprise the model.

7) The authors' could derive more constraints from the fly physiology literature than they do. As examples, Fisher et al., 2019 and Kim et al., 2019 have relevant findings relating to plasticity in mapping visual stimuli onto a compass representation. Turner-Evans et al., 2017 has a data-driven ring attractor model that is relevant, and Turner-Evans, 2019 features data demonstrating that the fly compass for current heading relies on visual input from the anterior optic tubercle, contrary to the authors' assumption deriving from an anatomical pathway from the posterior optic tubercle to the protocerebral bridge (175-176). On a somewhat related note, the fly heading system does not necessarily show 'bar following' in open loop: the experiments cited (Seelig and Jayaraman, 2015) were performed in closed loop, with the animal controlling bar position.

Thank you for this feedback. We have changed the neural pathway to generate the terrestrial heading (local compass) as suggested. See response to point 5.3.

8) The authors should also release and include an properly commented code that they used for modelling in the final submission.

Additional comments of the source code, the simulation implementations and a GUI have been uploaded through the full submission and also upload to Github repository: https://github.com/XuelongSun/InsectNavigationToolkitModelling.

9) Why is the velocity of the simulated ant (Vo = 1cm/s) so much slower than that of the real one (about 50cm/s)? This point must be discussed. Is there any fundamental reason?

Apologies there was a typo in our description of the model velocity that stated 1cm/s, but this should be 1cm/step. There is no link between the speed in the simulation and computation. Rather than simulated ant moves in 1cm steps, computes the direction move, takes a step and repeats. We have now corrected this in the text (Materials and methods section).

10) What would happen to the simulated ant if an obstacle was placed on the familiar route ? What is the robustness of the Zernike-based moment algorithm to the unpredicted presence of an obstacle that could appear during the homing ? Additional simulations to address this issue could show the robustness of the proposed navigation model. These new simulations could be in line with the well-known experiments proposed by Wehner and Wehner (R. Wehner and S. Wehner, “ Insect navigation : use of maps or Ariadne's thread?”).

This is a lovely idea and we have started to simulate such scenarios but believe that this is beyond the initial proof of concept nature of this study and better suited to a comparative study between this model and others across environments and with an investigation of parameters which will likely have a large effect on the performance. We have added a sentence to the Discussion raising exactly this idea as a logical next step in evaluating the model.

11) Subsection “ANN network and Route Following”: would it be possible to plot Crf with respect to angular orientation of the simulated ant in various place (every 10° steps for example).

We investigated plotting the data using crf as the reviewer suggested but as the preferred orientation changes with location this was very hard to visualise across locations. We found that this data was more intuitively presented using the quiver plots in the background of Figure 4B.

[Editors' note: further revisions were suggested prior to acceptance, as described below.]

Larger points for consideration (Optional revisions at the authors’ discretion):1) For increased accessibility and readability, please consider doing a little more with the earliest schematics to properly orient the broader readership. As an example, Figure 2A is fairly complicated for a reader who isn't familiar with the insect brain, and the panels at right will not be easy for most people to digest without Stone et al., 2017, open nearby (although the Stone et al. study is a must-read for anyone interested in insect navigation, this may not be the ideal way to get people to read the paper!). Could 'Shifted I-TB1' be unpacked a little by showing how the anatomy of the PB-FB columnar neurons might naturally facilitate the shift (as highlighted in the Stone et al. paper)-perhaps this could be a little breakout box at right. Note that the TB neurons should, in any case, be shown in the PB not the FB.

Thank you for the feedback. We have now revised Figure 2 to try and address the key issues. Specifically, we have replaced previous vector diagram and VH schematic with a combined schematic that we hope makes it easier to conceptually understand (a) how the steering circuit functions (b) how VH functions through a simple shifted heading input to the steering circuit. On reflection we think that if the readers understand these points then much of what follows should follow, and we think that these schematics should allow that.

Also, as requested we have relabelled what we previously called "shifted ITB1" neurons to VH as we can only speculate at this stage which neurons would store this signal.

Regarding adding detail to the shifting circuit. We considered this feedback and agree that it deserves addressing but we prefer to add a discussion of possible shifting mechanism to the text where we have space for some details and to add speculation.

2) Similarly, although the vector subtraction plots below are helpful to the informed reader, it is not clear that they would be sufficiently explanatory to a newer reader. Again, it is the authors' decision whether or not to do more here.

Finally, we have removed the vector plot entirely now as it has been superseded by the new Figure 2B and C.

3) A bigger, conceptual point: it is not obvious that one needs as many additional near-independent ring attractors as are invoked in this model, leaving aside the issue that they seem unlikely from an anatomical perspective. It is not clear or convincing that multiple ring attractors are needed to implement the authors' ideas. This potential opportunity for parsimony deserves some exploration, but the authors can decide whether that's something they want to do as part of this paper or not. The one thing that would be good is to make clear where the additional ring attractors reside in the authors' model: if they are speculatively placed in the FB, that should be made clearer in the early schematics (and also made clear that it is speculation at this stage).

We have added two paragraphs to the Discussion to address these points directly. Specifically, we clarify the benefits of the proposed RAs which may make them more parsimonious than alternate optimal integration networks. Also, we have clarified how many RAs we propose which we feel was confusing in previous edits. Finally, we add a call for future analysis of biological realism indicating that simpler mechanisms cannot be ruled out. We have included a label to the left of Figure 3A that indicates that we propose that the new integrating ring attractors reside in the FB. We have also added this clarification to the figure legend, and in the general discussion in the main text.